# Evolutionary lineage-specific genomic imprinting at the *ZNF791* locus

**Jinsoo Ahn**[1], **In-Sul Hwang**[2,3], **Mi-Ryung Park**[2], **Milca Rosa-Velazquez**[4¤], **In-Cheol Cho**[2], **Alejandro E. Relling**[4], **Seongsoo Hwang**[2], **Kichoon Lee**[1]*

**1** Department of Animal Sciences, The Ohio State University, Columbus, Ohio, United States of America, **2** National Institute of Animal Science, Rural Development Administration, Jeonbuk 55365, Republic of Korea, **3** Columbia Center for Translational Immunology, Columbia University Irving Medical Center, Columbia University, New York, New York, United States of America, **4** Department of Animal Sciences, The Ohio State University, Wooster, Ohio, United States of America

¤ Current address: Facultad de Medicina Veterinaria y Zootecnia, Universidad Veracruzana, Veracruz, Mexico

* lee.2626@osu.edu

## Abstract

Genomic imprinting is an epigenetic process that results in parent-of-origin effects on mammalian development and growth. Research on genomic imprinting in domesticated animals has lagged due to a primary focus on orthologs of mouse and human imprinted genes. This emphasis has limited the discovery of imprinted genes specific to livestock. To identify genomic imprinting in pigs, we generated parthenogenetic porcine embryos alongside biparental normal embryos, and then performed whole-genome bisulfite sequencing and RNA sequencing on these samples. In our analyses, we discovered a maternally methylated differentially methylated region within the orthologous *ZNF791* locus in pigs. Additionally, we identified both a major imprinted isoform of the *ZNF791-like* gene and an unannotated antisense transcript that has not been previously annotated. Importantly, our comparative analyses of the orthologous *ZNF791* gene in various eutherian mammals, including humans, non-human primates, rodents, artiodactyls, and dogs, revealed that this gene is subjected to genomic imprinting exclusively in domesticated animals, thereby highlighting lineage-specific imprinting. Furthermore, we explored the potential mechanisms behind the establishment of maternal DNA methylation imprints in porcine and bovine oocytes, supporting the notion that integration of transposable elements, active transcription, and histone modification may collectively contribute to the methylation of embedded intragenic CpG island promoters. Our findings convey fundamental insights into molecular and evolutionary aspects of livestock species-specific genomic imprinting and provide critical agricultural implications.

## Author summary

Genomic imprinting, an epigenetic process, influences mammalian development and growth through gene expression dependent on parental origin. Research on imprinting in domesticated animals has primarily focused on imprinted genes previously identified in mice and humans. However, we discovered that, unlike in mice and humans, the porcine

**Data Availability Statement:** WGBS and RNA-seq data generated in this study have been submitted to the NCBI Gene Expression Omnibus (GEO; https://www.ncbi.nlm.nih.gov/geo/) under accession number GSE263495 and GSE263494,

respectively. The publicly available datasets and VCF files used in this study are listed in S3 Table. Our published sheep RNA-seq data are available under accession number GSE253249.

**Funding:** This work was supported by the United States Department of Agriculture's National Institute of Food and Agriculture Hatch Grant (Project No. OHO01304, awarded to KL). The funders had no role in study design, data collection and analysis, decision to publish, or preparation of the manuscript.

**Competing interests:** The authors have declared that no competing interests exist.

*ZNF791* promoter is encompassed by a differentially methylated region with maternal methylation, and the *ZNF791-like* gene is expressed from only one allele in pigs, indicating novel imprinting of *ZNF791*. This imprinting appears to be conserved in the domesticated mammals we analyzed, including cattle, sheep, goats, horses, and dogs, but is absent in humans, non-human primates, and mice. Additionally, we provide mechanistic links between transcription and maternal methylation in porcine and bovine oocytes, highlighting the integration of transposable elements that leads to the methylation of intragenic CpG island promoters. Our findings suggest that genomic imprinting at the *ZNF791* locus has selectively evolved in livestock species.

## Introduction

Genomic imprinting is a primary epigenetic process that results in parent-of-origin-dependent gene expression in offspring. A subset of mammalian genes is affected by genomic imprinting, which fundamentally contributes to normal development and growth [1]. Epigenetic imprints, such as DNA methylation, are differentially established in the parental germ cells, i.e., oocytes and sperm. After fertilization, if these primary germline differentially methylated regions (gDMRs) resist genome-wide demethylation during pre-implantation development, they can serve as imprinting control regions (ICRs) [2]. Subsequently, ICRs dictate monoallelic expression depending on whether the allele is inherited from the mother or the father, leading to the formation of imprinted gene clusters in the majority of cases involving multiple coding and noncoding genes [3] or micro-imprinted domains in fewer cases in which surrounding genes escape imprinting [4].

The DMRs show spatially distinctive patterns. Sperm-derived gDMRs/ICRs are predominantly found in intergenic regions. In contrast, oocyte-derived gDMRs/ICRs, often located at CpG island (CGI) promoters, are associated with transcriptional repression that drives the silencing of the maternal allele and the expression of the paternal allele of corresponding genes [5]. Therefore, in this case of direct silencing, the paternal allele-specific expression is accompanied by a maternally methylated gDMR (hereafter, maternal gDMR) that originates from oocytes and is maintained throughout zygotic and post-implantation embryonic development. Since the discovery of genomic imprinting, molecular biology approaches have been applied to identify these gDMRs and the allelic expression of putative imprinted genes, utilizing single nucleotide polymorphisms (SNPs). It has been found that genomic imprinting occurs in therian mammals, including eutherians and marsupials, while there is no evidence of genomic imprinting in prototherians (monotremes), avians (birds), and lower vertebrates such as reptiles, amphibians, and fish [6].

Research on genomic imprinting in domesticated mammals has primarily targeted orthologous genes known to be imprinted in mice and humans [7]. This focus may restrict the identification of livestock-specific imprinted genes that do not have imprinted orthologs in mice and humans. Importantly, the generation of diploid parthenogenetic or gynogenetic embryos, which contain two maternal genomes in place of the paternal genome, leads to early lethality in mouse embryos, underscoring the necessity of both paternal and maternal genomes for normal embryonic development [8]. However, these types of uniparental embryos tend to survive longer in domesticated animals [9]. Based on this, our research focused on generating parthenogenetic porcine embryos and comparing them with biparental embryos to identify maternal imprints that have not been previously reported in mice and humans [10]. Utilizing these

embryos, we aim to investigate a maternal DMR located at a CGI promoter, alongside assessing concurrent differential gene expression patterns.

Specifically, analyzing imprinting states at the transcript level will allow us to gain detailed insights into the imprinting status in a locus. Further comparative analyses in other mammalian species, including other domesticated mammals, humans, non-human primates, and rodents, will serve to highlight the evolutionary conservation or non-conservation of imprinting, as well as explore potential mechanisms of *de novo* DNA methylation. To facilitate these locus-wise comparative analyses at the transcript level, we focused on orthologues within one of the largest gene families, the zinc finger protein gene family. This group provides a substantial number of gene members for analysis, given that less than 1% of genes are generally estimated to be subject to imprinting [11], and it includes at least one known imprinted gene, *PLAGL1* [12,13], in pigs, which serves as a positive control. Additionally, the imprinting statuses of porcine zinc finger protein genes have not yet been comprehensively studied, aside from their role in maintenance methylation [14].

In this study, we highlight the zinc finger protein 791-like (*ZNF791-like*) gene as an imprinted gene in pigs, which has not been extensively studied. Using whole-genome bisulfite sequencing (WGBS) and RNA sequencing (RNA-seq) on parthenogenetic and normal biparental porcine embryos, we identified a maternal DMR, an imprinted isoform, and a previously unannotated antisense transcript in pigs. Importantly, our comprehensive analyses of the orthologous genes across eutherian mammalian species revealed that the imprinting patterns were exclusive to domesticated mammals and absent in primates and rodents. Furthermore, we explored the potential mechanisms underlying the establishment of maternal imprints in oocytes, supporting the notion that a long terminal repeat (LTR) integration, upstream transcription initiation, and histone modification may act in concert to DNA methylation at embedded intragenic CpG island promoters. These findings describe lineage-specific *ZNF791* imprinting, offering insights into molecular and evolutionary aspects of epigenetic regulation among eutherian mammals.

## Results

### DNA methylation profiling and detection of differentially methylated regions

Following the generation of parthenogenetically activated (PA) and control (CN) porcine embryos as detailed in the Materials and Methods section (S1 Fig), we conducted WGBS with a mean sequencing depth of 44.6× and a mean genome coverage of 96.67% (S1 Table). Subsequently, we analyzed DNA methylation levels in these pig embryos within genomic regions (between –10 kb from the transcription start site [TSS] and +10 kb from the transcription end site [TES]) and putative promoter regions (defined as ± 2 kb from the TSS) of all 2,900 porcine zinc finger transcripts from 642 genes (S2 Table). Our analysis revealed that mean DNA methylation levels in the putative promoter regions were generally low (Fig 1A), indicative of a condition that is conducive to robust detection of DMRs in the promoter regions of these embryos. Moreover, methylation levels were higher in gene bodies (from TSS to TES) compared to flanking regions across embryos, aligning with the gene-body methylation pattern in mammals (Fig 1A). Based on the methylation ratios between PA and CN embryos, we initially selected the top 100 transcripts, with a minimum methylation ratio being approximately 1.2, and identified human orthologs for potential comparative analyses. Of these, the top 88 porcine zinc finger transcripts with orthologous loci in humans were highlighted (S2 Table). Further sorting based on the PA1 methylation levels revealed that, while methylation levels can be identical if the putative 4 kb promoter regions overlap and share CpG sites, the promoter

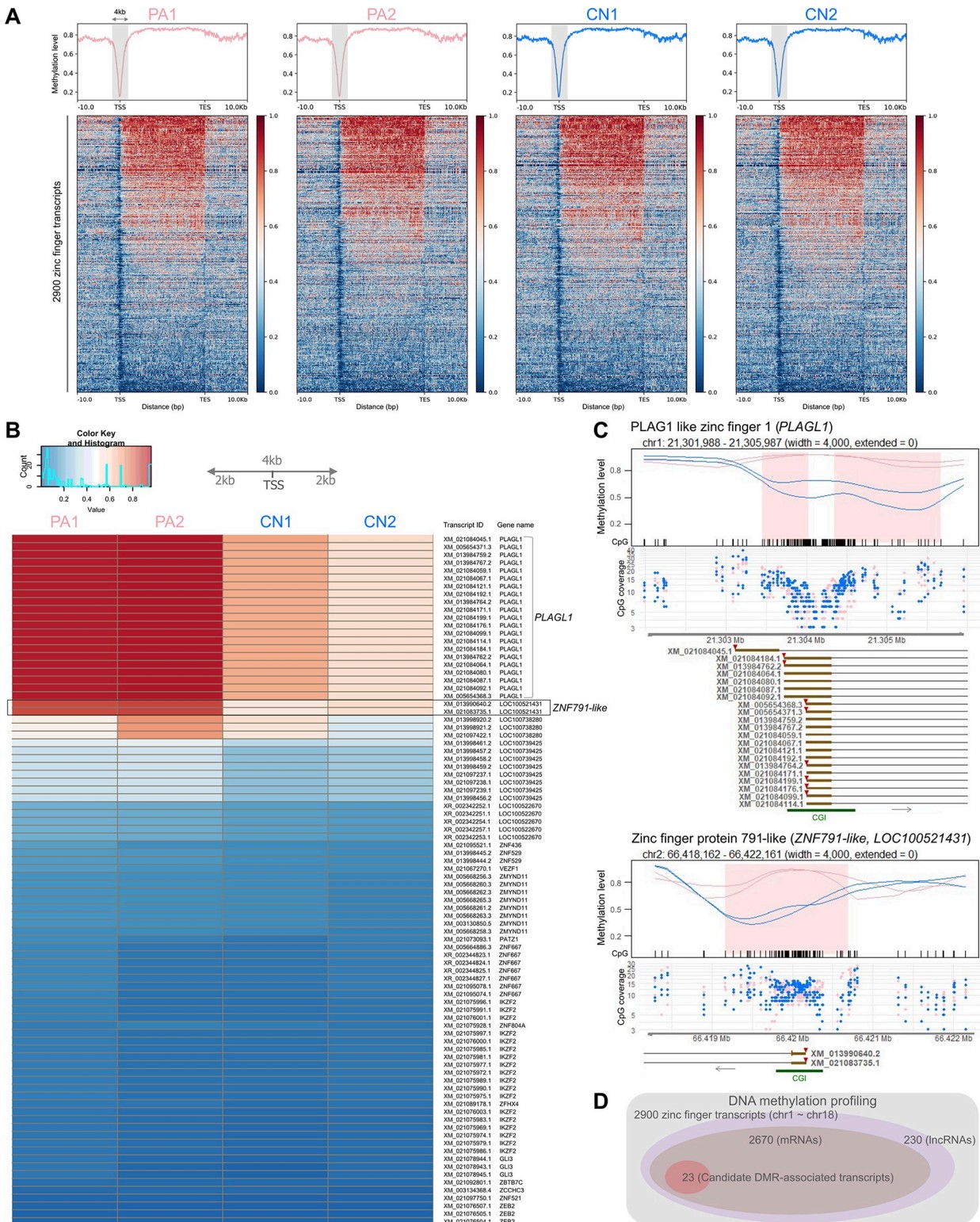

**Fig 1. Overview of imprinted transcript discovery. (A)** Mean DNA methylation levels of genomic regions in porcine embryos, including transcript bodies (TSS–TES) and ± 10 kb flanking regions. The ± 2 kb region from TSS is highlighted in grey. Methylation levels of 2,900 porcine zinc finger transcripts are presented with heatmaps in descending order. **(B)** A heatmap of mean CpG methylation levels within ± 2 kb of TSSs of selected transcripts. Based on methylation ratios between PA and CN embryos, the top 88 porcine transcripts with human orthologs were identified and re-sorted by methylation levels in PA1. **(C)** Smoothed methylation profiles for PA (pink) and CN (blue) replicates. BSmooth-

identified DMRs (pink-colored regions) for two genes are displayed, along with CpG sites, CpG coverages, transcripts, and CpG islands (CGIs). TSS positions are marked with red triangles on the first exons, with subsequent aligned transcripts sharing the same TSS not marked. Arrows indicate transcriptional directions. Mb, million base pairs (Mb). **(D)** From 2,900 porcine zinc finger transcripts, 23 mRNA transcripts from the *PLAGL1* and *ZNF791-like* genes were identified as candidate DMR-associated transcripts.

regions of 21 *PLAGL1* transcripts and two *ZNF791-like* transcripts exhibited high methylation in PA embryos and moderate methylation in CN embryos (Fig 1B and S2 Table). During the initial methylation profiling, this pattern suggested a maternal methylation effect, where PA embryos with only maternal alleles tended to have higher methylation levels than the control embryos.

We then employed a DMR detection software to identify differentially methylated regions between the PA and CN embryos within the putative promoter regions. Using the BSmooth method with default parameters, candidate DMRs were identified in the promoter regions (TSS ± 2 kb) with dense CpG sites, or CGIs, of two zinc finger protein genes, *PLAGL1* and *ZNF791-like* (Fig 1C). The individual CpG coverages were generally well above the minimum threshold of three (Fig 1C), and the mean CpG coverages ranged from 10.53 to 11.41 for XM_005654368.3 (*PLAGL1*) and from 11.15 to 11.69 for XM_013990640.2 (*ZNF791-like*) across samples (S2A and S2B Fig). The mean CpG coverages for all transcript isoforms are provided in S2 Table. Among 2,900 porcine zinc finger transcripts, there are 2,670 mRNAs and 230 long-noncoding RNAs (lncRNA), and 23 mRNA transcripts of *PLAGL1* and *ZNF791-like* were initially categorized as candidate transcripts that might be associated with DMRs in their putative promoter regions (Fig 1D). The known imprinted locus of *PLAGL1* was utilized as a positive control for the maternally methylated DMR. This DMR was further validated using the metilene software (FDR < 0.05) at the promoter region of expressed transcripts, and our RNA-seq analysis showed that the corresponding paternal expression pattern was exclusively observed in the control embryos for a subset of transcripts with the IDs XM_005654368.3, XM_005654371.3, and XM_021084067.1 (S2C Fig). Another DMR identified upstream by metilene was more than 3kb apart from the TSS of XM_021084156.1, falling outside the TSS ± 2 kb range (S2C Fig). In summary, differentially methylated regions were identified within the putative promoter regions of zinc finger transcripts at the porcine *ZNF791-like* locus, as well as at the *PLAGL1* locus, which served as a positive control.

## Locus-specific paternal expression of *ZNF791-like* was attributed to a major transcript isoform

Following the current annotation of pig reference genome assembly (Sscrofa11.1/susScr11), the *LOC100521431* (*ZNF791-like*) locus was explored to study the imprinting status. In the downstream (3′) of *MAN2B1* gene, short transcripts (XM_031990640.2 and XM_021083735.1) of *ZNF791-like* appeared to be expressed from the CN embryos exclusively that are initiated from a CGI (Fig 2A and 2B). In addition to this NCBI genome annotation, we integrated the *ENSSSCG00000033388* gene from the Ensembl genome annotation in the upstream (5′) of long transcripts of *ZNF791-like*. This Ensembl transcript is initiated from a CGI and appeared to be expressed from both PA and CN embryos. There also appeared to be an expression of previously unannotated antisense (AS) transcript overlapping the short transcripts of *ZNF791-like* (Fig 2A and 2B). Our profiling of DNA methylation status revealed a DMR spanning the CGI that regulates transcription of the short *ZNF791-like* transcripts (Fig 2C). This DMR resulted from full methylation in PA embryos and partial methylation in CN embryos, indicating maternal DNA methylation (i.e., maternal imprint) that could lead to silencing of the maternal allele of short *ZNF791-like* transcripts and their subsequent paternal expression.

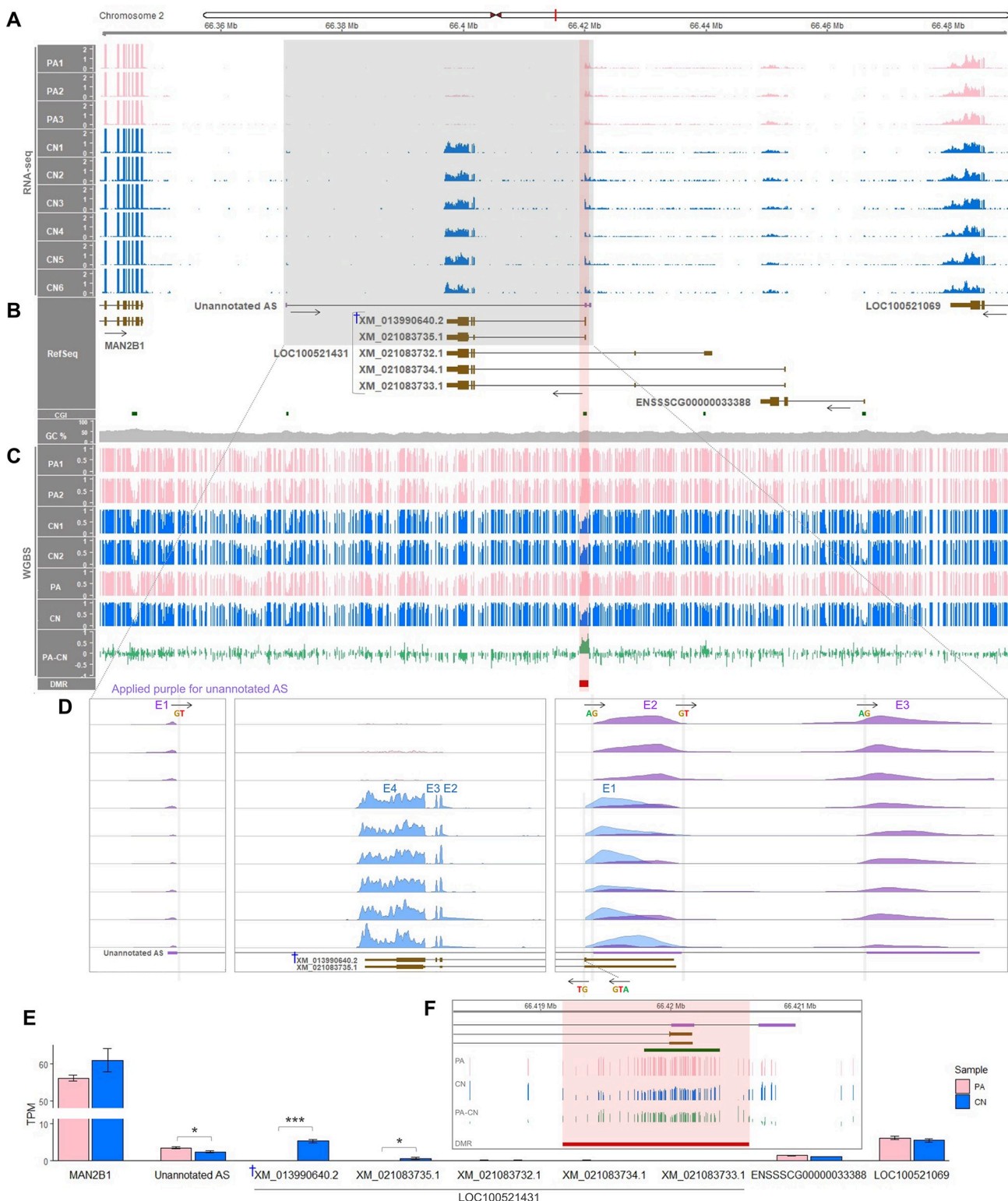

**Fig 2. Regional view of the *LOC100521431* (*ZNF791-like*) locus in pigs. (A)** RNA-seq read coverages of PA (pink, n = 3) and CN (blue, n = 6) embryos across a 15-Kb region (66.34–66.49 Mb), normalized to TPM values. **(B)** NCBI RefSeq annotation alongside an Ensembl-derived transcript (*ENSSSCG00000033388*) and an unannotated antisense (AS) transcript (purple). Protein-coding transcripts (brown) show translated regions as tall boxes and untranslated regions as short boxes. Arrows indicate transcriptional directions. CGI, CpG island; GC%, GC content. **(C)** WGBS DNA methylation levels (n = 2 for each of PA and CN embryos). Mean methylation ratios (PA and CN) are followed by their differences (PA-CN). A DMR (FDR < 0.05) is

marked in red. **(D)** Close-up view of expressed transcripts with exon (E) numbers. Splicing donor (GT) and acceptor (AG) are denoted with arrows for orientation. The start codon (ATG) marks the beginning of a protein-coding open reading frame. **(E)** Quantification of expressed transcripts and genes. TPM values are represented as mean ± SEM. *, $P < 0.05$; ***, $P < 0.001$; †, a major predominant *ZNF791-like* transcript. **(F)** Zoom-in of the mean methylation levels at CpG resolution, with the DMR highlighted in red.

In contrast, the CGI regulating transcription of the *ENSSSCG00000033388* gene was unmethylated in both PA and CN embryos (Fig 2C).

A zoomed visualization of the short *ZNF791-like* transcripts showed prominent exon 3 (E3) usage, instead of E3 skipping, as seen in the transcript XM_031990640.2 (Fig 2D). A combined close view revealed that exon 2 (E2) of the unannotated AS overlapped with exon 1 (E1) of the short *ZNF791-like* transcript (Fig 2D). The splicing of the unannotated AS was supported by the presence of splicing donor (GT) and accepter (AG) sites, allowing the inference of its transcriptional direction based on the orientation of the GT and AG sequences (Figs 2D and S3). Of note, expression of the unannotated AS tended to be higher in PA embryos than in CN embryos. Further quantification of transcript expression revealed that the mean expression level of the major short *ZNF791-like* transcript (XM_031990640.2) in the CN embryos was approximately 8-fold higher than that of another short transcript (XM_021083735.1): 5.36 ± 0.40 TPM for XM_031990640.2 and 0.66 ± 0.23 TPM for XM_021083735.1 (Fig 2E). Notably, expression of the unannotated AS was significantly 1.43-fold higher in the PA embryos than in the CN embryos ($P < 0.05$): 2.43 ± 0.22 TPM in CN and 3.49 ± 0.25 in PA embryos (Fig 2E), suggesting a potential maternally biased expression. This could further support the *ZNF791-like* imprinting and paternal expression, which might affect the expression of a nearby overlapping gene. The methylation levels at single CpG resolution highlighted the imprinted DMR, which encompasses the first exons and the CGI (Fig 2F). In short, the paternal expression of *ZNF791-like* was greatly due to one of the short transcripts (XM_031990640.2), that could be regulated by the DMR, and potentially led to greater expression of the overlapping unannotated AS transcript in the PA embryos than in the CN embryos.

## DNA methylation imprint in the *ZNF791* locus was species-specific

To examine conservation of *ZNF791* imprinting in mammalian species, we analyzed publicly available WGBS data (listed in S3 Table) from gametes, tissues, and cells of 13 species. In humans, mice, and rats, the *ZNF791* or *Zfp791* promoter regions were unmethylated or significantly hypomethylated in both oocytes and sperm, indicating a lack of imprinted gametic methylation (Fig 3A). On the other hand, in pigs and cows, the promoter CGIs were fully methylated in oocytes and unmethylated in sperm, suggesting a conserved pattern of imprinted gametic methylation (Fig 3A). To further investigate maintenance of methylation patterns, various tissues and cells were analyzed. In humans, non-human primates (crab-eating macaque, chimpanzee, rhesus monkey, and gibbon), and rodents (mice and rats), all the CGI promoters or the promoter regions were unmethylated or significantly hypomethylated in fetal and adult tissues, including f-Br (fetal brain), Br (brain), f-Mu (fetal muscle), Mu (muscle), He (heart), Li (liver), Lu (lung), Ki (kidney), and Bl (blood) (Fig 3B).

To the contrary, in domesticated mammals (pigs, cows, goat, sheep, horses, and dogs), all the CGI promoters were partially methylated in fetal and adult tissues and cells, including Fb (fibroblast), f-Li (fetal liver), y-Mu (young skeletal muscle, day 40), a-Mu (adult skeletal muscle, day 180), a-Sn (angen stage of skin), t-Sn (telogen stage of skin), e-Mu (embryonic muscle, embryonic day 110), a-Mu (adult skeletal muscle, 2-years-old), as well as some tissues that share the same abbreviation as those listed above (Fig 3B). Whether the partial methylation is

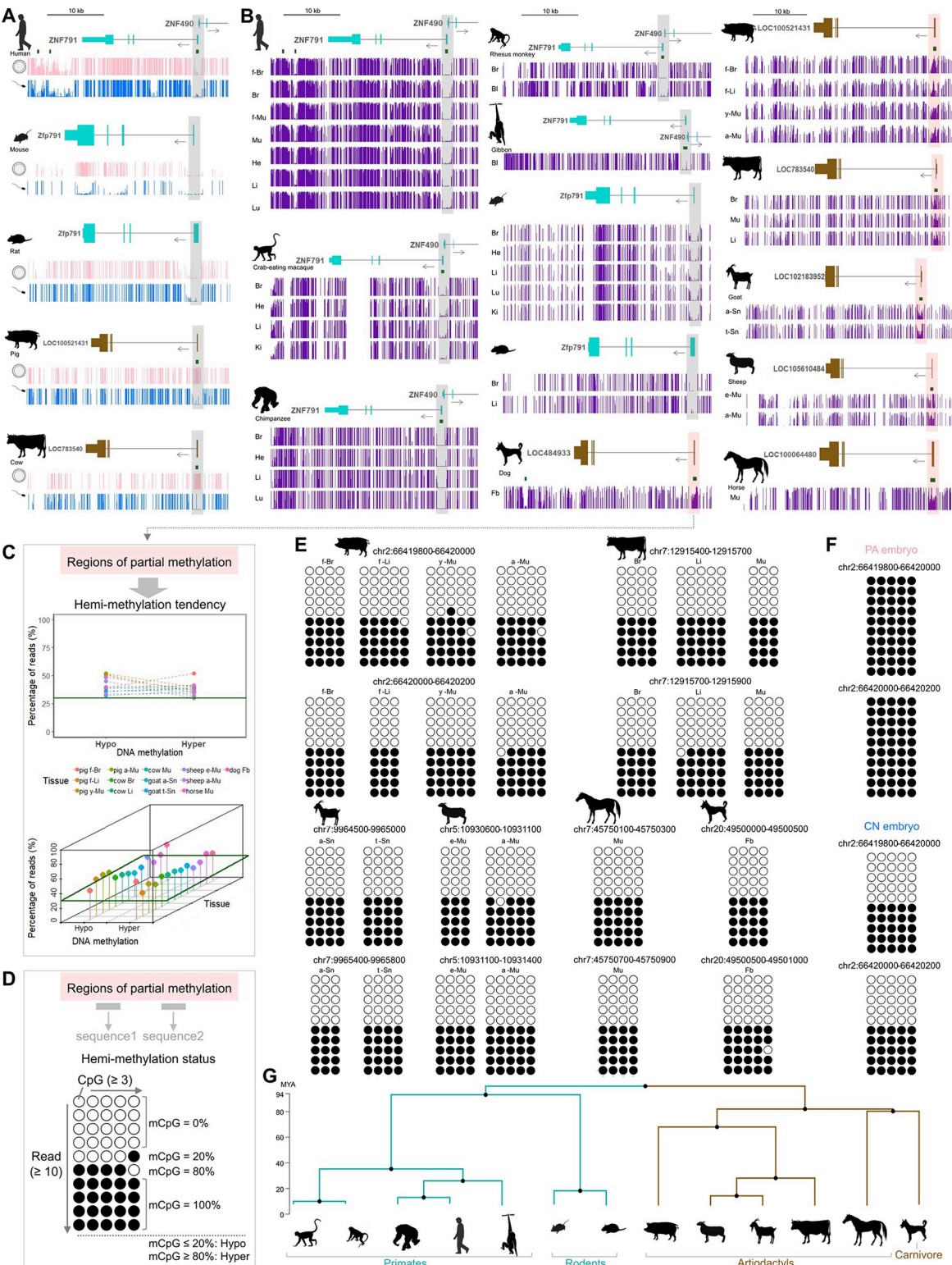

**Fig 3. DNA methylation status of the *ZNF791* locus in mammalian cells and tissues. (A)** Methylation levels (0 = unmethylated, 1 = fully methylated) in oocytes (pink) and sperm (blue), with promoter regions of *ZNF791* (or *Zfp791*) and *ZNF791-like* highlighted in grey, alongside CpG islands (green). **(B)** Methylation levels in tissues and cells (e.g., f-Br, fetal brain; see abbreviations in Results), with transcripts color-coded cyan for primates and rodents, and brown for other mammals. Partially methylated promoter regions are shaded in pink. **(C)** Analyses of partial methylation. WGBS reads were extracted and analyzed for hemi-methylation, applying a threshold of at

least 30% of hypomethylated and hypermethylated reads defined in D. **(D)** Hemi-methylation status evaluated in two sequences with ≥ 10 reads and ≥ 3 consecutive CpG dinucleotides per read. The percentage of mCpGs was then calculated (see Materials and Methods). **(E)** Evaluation of hemi-methylation status across species. WGBS reads were extracted from indicated regions. Open and closed circles indicate unmethylated and methylated CpGs, respectively. **(F)** Full- and hemi-methylation states in PA and CN embryos, respectively. **(G)** A phylogenetic tree showing divergence. MYA, million years ago.

originated from methylation difference between parental alleles (i.e., hemi-methylation pattern) could be analyzed based on WGBS reads. All reads overlapping the partially methylated regions tended to be either hypomethylated or hypermethylated, while the full methylated region in PA embryos showed hypermethylation only (Figs 3C and S4–S10 and S4 Table). Our definition of hypo- or hypermethylation is the percentage of methylated CpG dinucleotides (mCpGs) less than 20% or more than 80% and is consistent with previous approaches [15] (Fig 3D). In at least two consecutive CpG dinucleotide sequences within reads overlapping the partially methylated regions, the percentage of mCpGs were either 0 ~ 20% (hypomethylation) or 80 ~ 100% (hypermethylation) in analyzed individual reads, indicating hemi-methylation status similar to that in CN embryos, whereas PA embryos showed full methylation (Fig 3E and 3F). Estimated divergence time between clade 1 (primates and rodents) and clade 2 (artiodactyls and carnivore) was *circa* 94 million years ago (MYA) (Fig 3G). Consequently, DNA methylation imprints in the *ZNF791* or *ZNF791-like* locus appeared in one group of the analyzed mammalian species (artiodactyls and carnivore), but not in another group (primates and rodents), indicating a species-specific imprinting pattern.

## Bi or mono-allelic expression of the *ZNF791* gene across species based on individual-matched genomic DNA and RNA-seq reads

With the aim of investigating whether *ZNF791* expression is non-imprinted or imprinted, its biallelic or monoallelic expression was to be examined across mammalian species. To achieve this, we first identified single nucleotide polymorphisms (SNPs) where two different alleles exist (i.e., heterozygous or informative SNPs), except for inbred mice. In humans, rhesus monkeys, chimpanzees, dogs, pigs, and cows, heterozygous SNPs were identified on genomic DNAs covering exons of *ZNF791* (Fig 4 and S5 Table). Whether *ZNF791* is expressed from both alleles or from a single allele was determined by analyzing the expression of alleles at the SNP sites in various normal tissues and cells. In RNA-seq data derived from the same individual whose genomic DNA was sequenced, both alleles were expressed in humans, rhesus monkeys, and chimpanzees in analyzed transcripts (Figs 4, S11 and S12, and S5 Table). On the other hand, in dogs, pigs, and cows, either a single alternative or reference allele was expressed in predominantly expressed isoforms in all analyzed tissues (Figs 4 and S13, and S5 Table). Additionally, the unannotated AS was expressed in the liver of a 60-day-old pig (S13C Fig). In stranded RNA-seq data from the liver of 180-day-old pigs, the transcriptional direction of the unannotated AS transcript was opposite to that of *ZNF791-like* (S13D and S13E Fig). In contrast, due to the presence of homozygous alleles at each SNP (i.e., homozygous SNPs) in inbred mice, a different approach was applied for the mouse analysis. It revealed that offspring from both initial and reciprocal crosses between CAST/EiJ and C57BL/6J strains consistently expressed both alleles in expressed transcripts (Figs 4 and S12D, and S5 Table). In summary, *ZNF791* was found to be expressed biallelically in humans, non-human primates, and mice, indicating a non-imprinted expression in these species. Conversely, in dogs, pigs, and cows, *ZNF791* or *ZNF791-like* exhibited mono-allelic expression at the heterozygous SNP sites, indicative of an imprinted expression in these mammalian species.

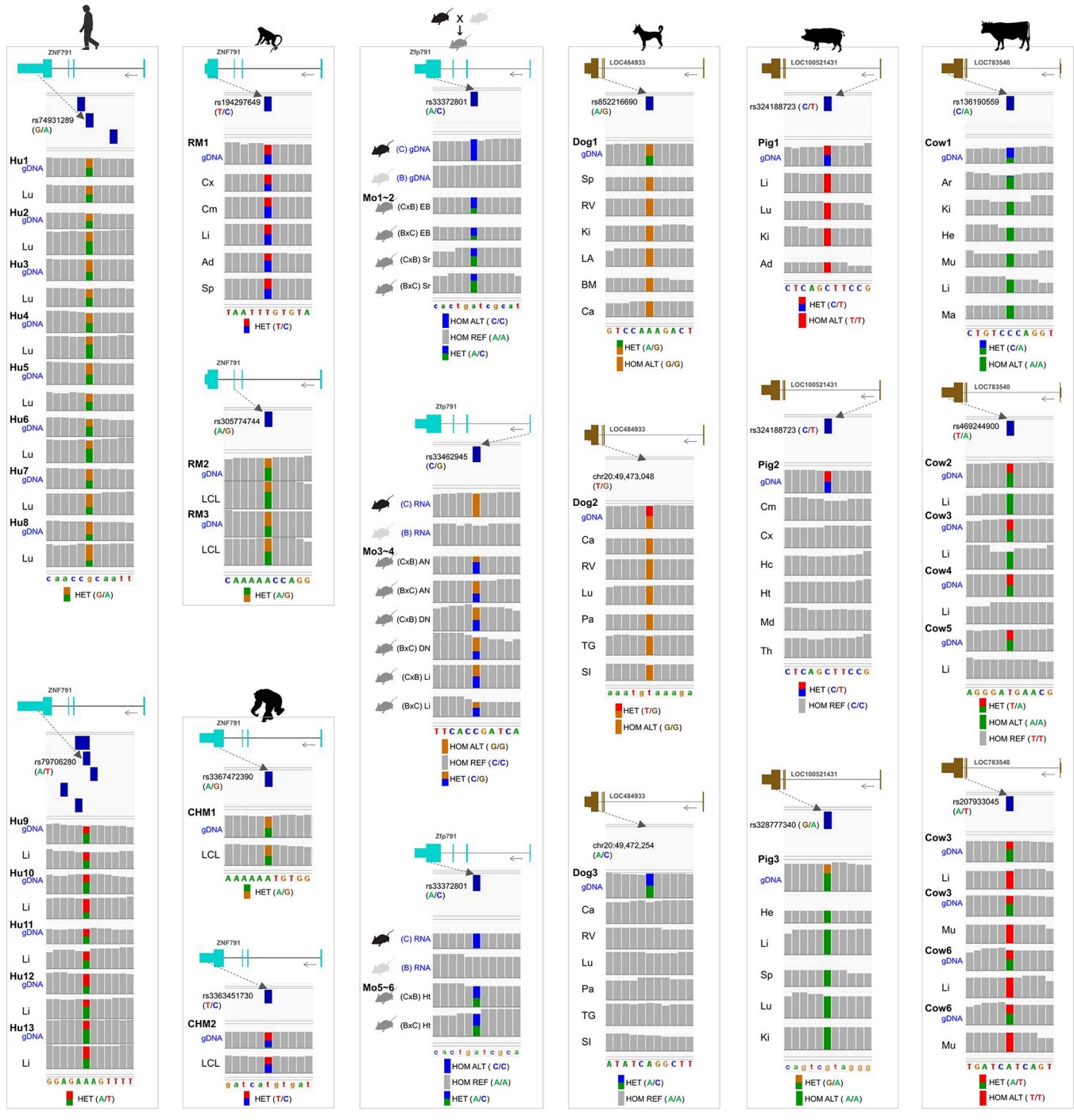

**Fig 4. Bi- or monoallelic expression of *ZNF791* or *ZNF791-like* across mammalian species.** i) Whole genome/exome sequencing data was used to identify heterozygous SNPs for analyzing bi- or monoallelic mRNA expression. For inbred mouse strains, homozygous SNPs were chosen to analyze allelic expression in offspring from reciprocal crosses. ii) In the same individuals, mRNA expressions in normal tissues or cells were analyzed to determine allelic expression patterns. Reference SNP identifiers (rs IDs) are accompanied by reference/alternative alleles ('ref/alt'). For SNPs without rs IDs, genomic coordinates and alleles are specified. Hu, humans; RM, rhesus monkey; CHM, chimpanzee; Mo, mouse with reciprocal crosses of CAST/EiJ (C) and C58BL/6J (B). Tissues: Lu, lung; Li, liver; Cx, cortex; Cm, cerebellum; Ad, adipose tissue; Sp, spleen; LCL, lymphoblastoid cell line; EB, embryonic brain; Sr, striatum; AN, arcuate nucleus; DN, dorsal raphe nucleus; RV, right ventricle; Ki, kidney; LA, left atrium; BM, bone marrow; Ca, cartilage; Pa, pancreas; TG, thyroid gland; SI, small intestine; Hc, hippocampus; Md, midbrain; Th, thalamus; He, heart; Ar, adrenal cortex; Mu, skeletal muscle; Ma, mammary gland. Accession numbers and ID are listed in S5 Table. Alleles: HET, heterozygous; HOM REF, homozygous reference; HOM ALT, homozygous alternative.

## LTR-mediated transcription might regulate *ZNF791* imprinting in pigs and cattle

To investigate potential mechanisms behind the imprinting of the maternal allele of *ZNF791* gene, we analyzed transcription in oocytes and histone modifications during early development. Because the CGI promoter of *ZNF791* was maternally imprinted (Figs 2 and 3), there could be oocyte-specific events that contribute DNA methylation. In fully grown porcine oocytes, the *ZNF791-like* transcript (NCBI accession no. XM_021083732.1) was expressed, initiating from the TSS located near another putative CGI promoter and the LTR sequence, LTR52 (Fig 5A and 5B, grey shading, S14 and S15). This LTR52 was identified through a search in the Dfam database (E-value = $3.7 \times 10^{-5}$), which uses profile hidden Markov models (HMMs) to improve the sensitivity of detection (S16 Fig). Additionally, H3K4me3, a hallmark of promoters, was found enriched in the CGI promoter region of this transcript XM_021083732.1 (Fig 5C, grey shading). From the site where H3K4me3 enrichment ended, both H3K36me2 and H3K36me3 tended to be enriched, supporting their role in *de novo* gene-body methylation in oocytes (Fig 5C and 5D) [16,17]. During zygotic development (4 cell, 8 cell, and blastocyst), H3K4me3 became enriched around the intragenic CGI at the blastocyst stage, indicating transcriptional permissiveness of the transcripts XM_013990640.2 and XM_021083735.1 (Fig 5B, †, and 5C, red shading).

In bovine oocytes, the *ZNF791* transcripts (XM_015471970.2, XM_015471971.2, and XM_024994880.1) were expressed, initiating from TSSs located near a putative CGI promoter and an LTR52 element (E-value = $1.2 \times 10^{-11}$) (Fig 5E and 5F, grey shading, and S16−S17). Enrichment of H3K4me3 further supported the active transcription of those transcripts (Fig 5G, grey shading). From the site where H3K4me3 terminated, H3K36me2/me3 enrichment began in oocytes, indicating their role in *de novo* gene-body DNA methylation (Fig 5G and 5H). The demarcation of H3K36me2/me3 (i.e., the formation of a boundary) was somewhat more pronounced in bovine oocytes compared to porcine oocytes. As zygotes developed, H3K4me3 became enriched at the intragenic CGI during the blastocyst stage (Fig 5G, red shading).

Overall, the patterns of transcription initiation, histone modification, and DNA methylation were similar between porcine and bovine oocytes and developing zygotes. Taken together, these data suggest a mechanism whereby transcription was initiated near the LTR element during oogenesis, leading to the recruitment of DNA methyltransferase, subsequent *de novo* gene-body DNA methylation at the intragenic CGI, and the establishment of the DMR.

## The *ZNF791* locus in humans and mice lacks genomic imprinting

The imprinting mechanism described above was previously identified by Bogutz et al. (2019) [18] for the *Impact* gene (S18 Fig) and the *Slc38a4* gene, both of which are imprinted in mice. The imprinted expression of *Impact* has also been reported in rats [19], but not in humans and pigs, indicating species-specific imprinting patterns. In contrast, the *ZNF791* (or *Zfp791*) gene exhibited no maternal imprints in both humans and mice (Fig 3) and showed non-monoallelic expression in both species (Fig 4), indicating a non-imprinted status. To confirm the absence of genomic imprinting at this locus, we further explored the *ZNF791* (or *Zfp791*) region. In human oocytes, the only annotated transcript of *ZNF791* was expressed, to a lesser degree than the *ZNF490* transcript, which is transcribed in the opposite direction (Fig 6A and 6B). Also, H3K4me3 was consistently enriched in the CGI promoter region during both gametic and zygotic stages (Fig 6C, grey shading). The absence of an additional CGI promoter in the downstream region suggests the lack of canonical maternal imprints in human oocytes (Fig 6D).

In mouse oocytes, only one *Zfp791* transcript was expressed, with no other transcripts annotated to be transcribed from the downstream region (Fig 6E and 6F). The promoter

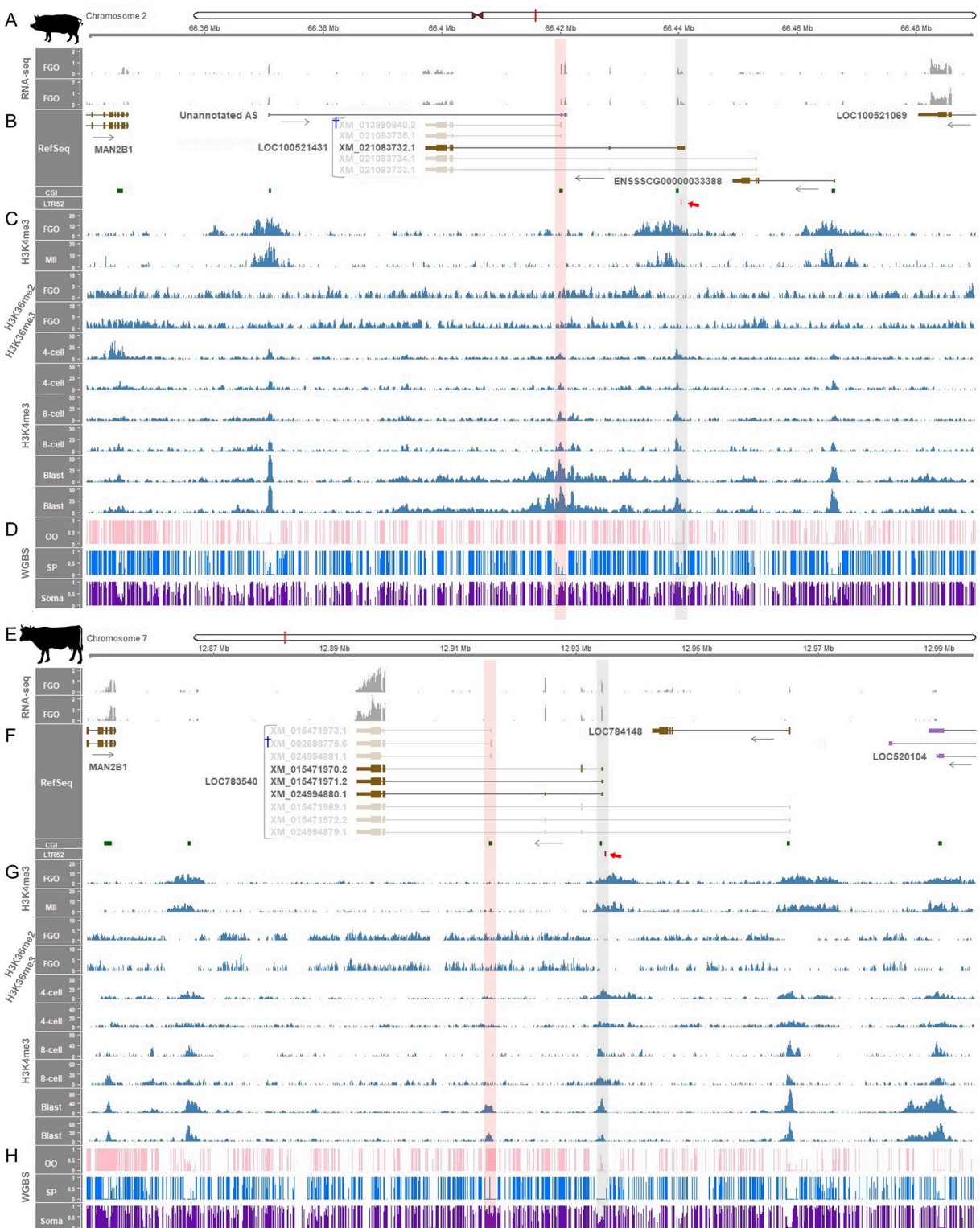

**Fig 5. Establishment of methylation imprints in porcine and bovine oocytes. (A, B)** RNA-seq read coverages (TPM) from porcine fully grown oocytes (FGOs) and the expressed *ZNF791-like* transcript. CGI and LTR52, near the TSS (chr2:66,440,912), are marked with green and red rectangles, respectively, and highlighted with grey shading. LTR52 is further indicated by a red arrow. Non-expressed transcripts are faded. **(C)** Enrichment of histone modifications, H3K4me3 and H3K36me2/me3, during porcine pre-implantation development. **(D)** WGBS methylation ratios: Full methylation in porcine oocytes (OO), unmethylation in sperm (SP), and partial methylation in somatic tissue (d120

muscle), shaded red. **(E, F)** RNA-seq coverages from bovine FGOs and expressed *ZNF791* transcripts. CGI and LTR52 near the TSS (chr7:12,934,209) are marked and shaded as in B. Faded are non-expressed transcripts. **(G)** Histone modifications, H3K4me3 and H3K36me2/me3, during bovine pre-implantation development. **(H)** WGBS methylation ratios of bovine samples: Full methylation (OO), unmethylation (SP), and partial methylation (somatic tissue; muscle), shaded red. Stages: MII (MII oocytes), 4-cell, 8-cell, Blast, (blastocysts). From the red-shaded region, the major predominant *ZNF791* transcripts, marked with † in (B) and (F), are preferentially transcribed at later developmental stages.

region was enriched with H3K4me3 histone modification (Fig 6G, grey shading). Enrichment of H3K36me3 occurred in the gene body of *Zfp791* (Fig 6G). The absence of a downstream CGI promoter region and additional annotated transcripts suggests a deficiency of canonical maternal imprints in mouse oocytes (Fig 6G), in contrast to the mouse *Impact* gene, which contains the canonical maternal imprint and exhibits partial methylation in somatic tissue (S18 Fig).

Unlike in pigs and cattle, maternal methylation imprints were not established in human and mouse oocytes, attributable to the absence of additional CGI promoters and downstream *ZNF791* transcript isoforms. This suggests that divergence between these domesticated animals and humans and mice has led to lineage-specific imprinting at the *ZNF791* locus.

## *ZNF791* imprinting implies selective forces of LTRs and evolution of lineage-specific imprinting

To further investigate orthologous imprinted *ZNF791* loci, we first compared the loci and LTRs in artiodactyls and dogs. Focusing on the *ZNF791* sequences immediately downstream (3′) of the *MAN2B1* gene in each species, we found that LTR52-derived sequences are present near the second CGI in domesticated Artiodactyla species (pigs, cows, sheep, horses, and goat), while an LTR103-derived sequence is present near the second CGI in dogs (order Carnivora) (Fig 7A, grey shaded rectangles). E-values, or expectation values, for LTRs derived from the Dfam database were as follows: pig LTR52 (E-value = $3.7 \times 10^{-5}$), cattle LTR52 (E-value = $1.2 \times 10^{-11}$), sheep LTR52 (E-value = $6.8 \times 10^{-6}$), horse LTR52 (E-value = $2.4 \times 10^{-6}$), goat LTR52-int (E-value = $7.0 \times 10^{-5}$), and dog LTR103_Mam (E-value = 0.051). These values indicate significant sequence similarity with the consensus sequences of LTR52 or LTR52-int (E-value < 0.05) and suggest a potential similarity with the consensus LTR103_Mam sequence in dogs. As imprinted *ZNF791* transcripts are expected to be expressed from the third CGI downstream (3′) of LTRs (Fig 7A, red shaded rectangles) in pigs and cows (Figs 2 and 5, and S13B and S13C), in dogs (S13A Fig), and in sheep, horses, and goats (S19A and S19B Fig), it suggests that the conservation of *ZNF791* imprinting in the analyzed species is also subject to an analogous pattern. The phylogenetic analysis of domesticated animals showed that the genetic distances between LTR52 sequences and the consensus LTR52 sequence are smaller than the distances between goat- and dog-derived sequences and the same consensus sequences (Figs 7B and S20). From the motif discovery/comparison and scanning, distribution of transcription factor binding sites (TFBSs) in LTR52 and its derivatives were identified (Fig 7C). From the motif discovery and comparison, DNA binding motifs of E2F3 and E2F2 (E2F2_DBD_1 and E2F3_DBD_1) were found to be significant in LTR52 sequences from pigs, cows, sheep, and horses (E-value < 0.05, *q*-value < 0.05) (Figs 7C, 7D and S21). Also, from motif scanning, binding sites for the MYBL2 transcription factor were found in the LTR52 sequences from pigs, cows, sheep, horses, and goat_LTR52-int (*q*-value < 0.05) (Fig 7C and 7D). Binding sites for the ZNF770 transcription factor were found in the LTR52 sequences from pigs, cows, sheep, horses, and dog_LTR103_Mam (*q*-value < 0.05) (Fig 7C and 7D).

The phylogenetic time tree suggests that the insertion of the *ZNF791* gene likely occurred around 94 MYA (Fig 7E, *β*), although the immediate downstream gene, *MAN2B1*, predates the

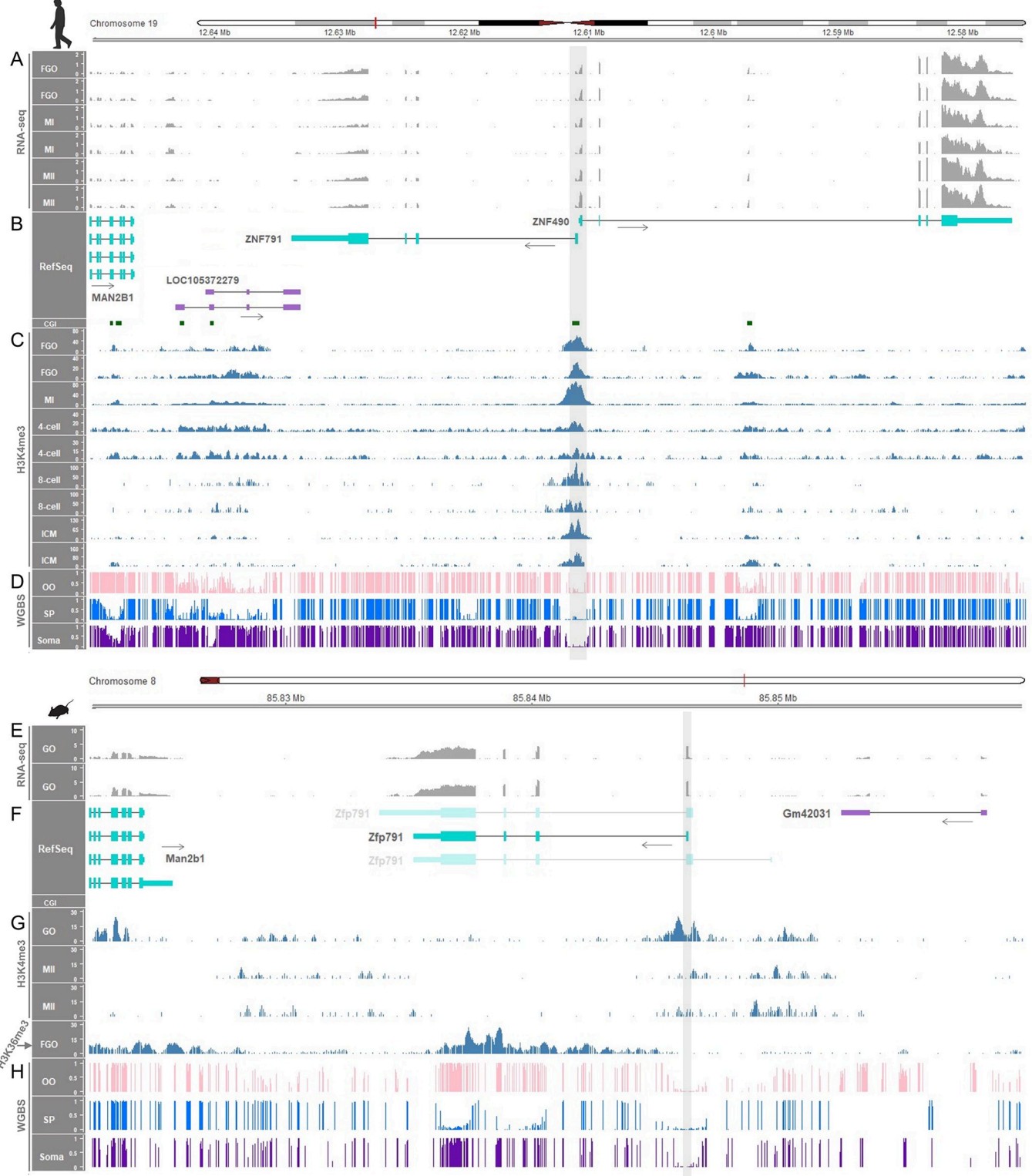

**Fig 6. Non-establishment of methylation imprints in human and mouse oocytes.** (**A, B**) RNA-seq read coverages (in TPM) from human oocytes at FGO, MI, and MII stages and the expressed *ZNF791* transcript. (**C**) H3K4me3 histone modifications during developmental stages of oocytes and zygotes. ICM, inner cell mass of the blastocyst. (**D**) WGBS methylation ratios of human samples. Unmethylation in oocytes (OO), sperm (SP), and somatic tissue (adult adipose tissue) are highlighted with grey shading. (**E, F**) RNA-seq read coverages from mouse growing oocytes (GOs) and the expressed *Zfp791* transcript. Non-expressed transcripts are presented as faded. (**G**) Histone modifications, H3K4me3 and H3K36me3, in mouse oocytes. (**H**) WGBS methylation ratios of mouse samples. Unmethylation in OO, SP, and somatic tissue (kidney) are highlighted with grey shading. All mouse samples are from the C57BL/6 strain.

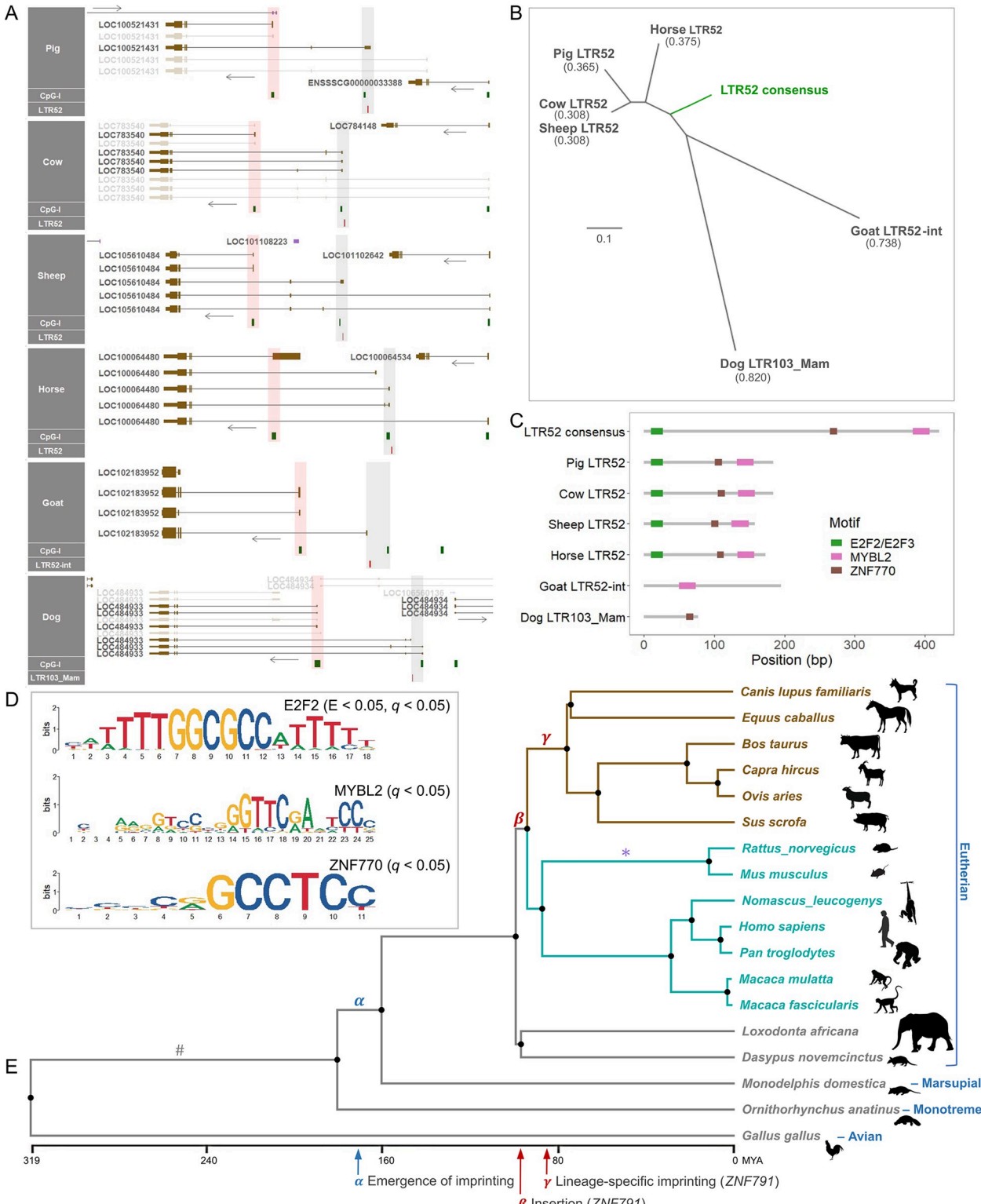

**Fig 7. LTR52 sequence comparison across the *ZNF791* genomic regions in domesticated animals. (A)** LTR locations near CGIs are marked by red rectangles in grey shaded areas. Red shading indicates the 1st exon of candidate imprinted transcripts and CGI promoter regions. **(B)** Phylogenetic analyses of genetic distances between LTR sequences. The scale bar represents evolutionary distance, with a value of 0.1 indicating a substitution per nucleotide, as estimated by the Kimura 2-parameter (K2P) model. **(C)** Transcription factor binding motifs detected in LTR sequences. **(D)** Logos of the significant motifs. **(E)** Phylogenetic time tree with artiodactyls and carnivores highlighted in brown, and primates and

rodents in cyan. Species silhouettes are from phylopic.org. Potential evolutionary events are marked with letters and arrows. MYA, million years ago; #, inclusion of the *MAN2B1* gene; *, lineage-specific imprinting (*Impact*).

emergence of genomic imprinting (Fig 7E, #). In *Gallus gallus* (chicken, avian), the whole locus containing both the *MAN2B1* gene and the *ZNF791* gene is missing from the NCBI RefSeq database. In *Ornithorhynchus anatinus* (platypus, monotreme), *Monodelphis domestica* (gray short-tailed opossum, marsupial), *Dasypus novemcinctus* (nine-banded armadillo, edentate), and *Loxodonta africana* (African savanna elephant, proboscidean), the *MAN2B1* gene exists but the downstream *ZNF791* gene is absent. In addition to the proposed emergence of imprinting at least 160 MYA following the divergence between monotremes and therians (marsupial and eutherian mammals) (Fig 7E, $\alpha$), lineage-specific imprinting of *ZNF791* might have occurred at least approximately 78 MYA in the common ancestors of artiodactyls and dogs (Fig 7E, $\gamma$). This lineage-specific imprinting might have involved the acquisition of upstream LTRs, overlapping transcripts, and the maternal gDMR locus, as illustrated by the *ZNF791* locus in this study. Similarly, lineage-specific imprinting of *Impact* may have occurred relatively recently in the common ancestors of rodents, involving the acquisition of maternal gDMR (Fig 7E, *, and S18) [18,19]. The *ZNF791* imprinting also leads to imprinted maternal expression of the overlapping unannotated AS in pigs (Fig 2D), and based on the pig QTLdb, a SNP located 327 bp upstream of the 1st exon of the unannotated AS is associated with residual feed intake (RFI) at a suggestive level ($P = 2.89E-05$) (S22 Fig), Taken together, in analyzed domesticated mammals (pigs, cows, sheep, horses, goat, and dogs), genomic imprinting at the *ZNF791* or *ZNF791-like* locus might have been selectively evolved with LTR sequence integration.

## Discussion

Herein, we conducted a comparative investigation of the imprinting statuses of *ZNF791* across 13 mammalian genomes, focusing on previously unreported DMRs, unannotated transcripts, and several key factors: CpG island promoter methylation, allele-specific expression, transcription in oocytes, integration of transposable elements, and lineage-specific imprinting. To identify candidate maternal imprints, we initially compared DNA methylation levels between bimaternal parthenogenetic embryos and biparental control embryos from pigs, utilizing two biological replicates from WGBS with sequencing depths ranging from 43.8× to 45.2× (S1 Table) that met the ENCODE standards, which require a minimum of two biological replicates with at least 30× coverage for reproducibility (see *Standards and Guidelines for Whole Genome Shotgun Bisulfite Sequencing (WGBS)* under the 'standards document' section at ENCODE project; https://www.encodeproject.org/documents/). This comparison facilitated the robust detection of maternal DMRs at CGI promoters which implicated transcriptional repression of the maternal allele [5]. Subsequently, to determine whether the maternal DMR detected in the *ZNF791-like* locus constitutes a germline DMR, we analyzed WGBS methylation levels from oocytes and sperm. Interestingly, the CGI promoters exhibited full methylation in oocytes from pigs and cattle, yet remained unmethylated in oocytes from humans, mice, and rats, indicating species-specific maternal gDMR imprints. The maintenance of allele-specific DNA methylation in somatic tissues of outbred animals were analyzed by examining methylation levels of consecutive CpG dinucleotides in individual WGBS reads (a read-based method), independent of SNPs [15]. It revealed that hemi-methylation (i.e., reflection of full methylation in one of the parental alleles and unmethylation in the other) at the CGI promoters is observed only in artiodactyls and dogs. It suggests conservation of this maternal gDMR in domestic

animals and led us to examine allele-specific gene expression to identify corresponding imprinted genes.

Genomic imprinting can affect all transcript isoforms in an isoform-independent manner, however when there are multiple promoters and they are imprinted in different ways, a mixed pattern of imprinting can occur (isoform-dependent imprinting) [20]. In this study, based on our methylation profiling which revealed maternal DMRs and suggested corresponding paternal gene expression, we included more control samples for RNA-seq to clearly show the paternal expression pattern, if present, in the CN group, using a different set of whole embryos having unique identifying numbers compared to those used for WGBS. Among previously annotated transcripts in the NCBI RefSeq database, we identified the major *ZNF791* transcript isoforms that were predominantly expressed in pig CN embryos, as well as in adult dogs, pigs, cattle, sheep, horses, and goats, indicating that *ZNF791* undergoes isoform-independent imprinting. Accordingly, we revealed monoallelic expression of *ZNF791* in these domestic animals, contrasting with the biallelic expression observed in humans, non-human primates, and mice. Additionally, the presence of long non-coding RNAs (lncRNAs) within an imprinted locus can further complicate the imprinting landscape, especially when these lncRNAs overlap with protein-coding mRNA transcripts and run in an antisense orientation. This overlap can lead to allele-specific silencing of mRNAs and ncRNAs through two possible mechanisms: i) transcriptional interference of a promoter or an enhancer and ii) RNA-interference (RNAi) [3]. The former mechanism, transcriptional interference, was demonstrated by the paternal allele-specific silencing of *Igf2r*, which requires the overlap of *Airn* lncRNA transcription over the *Igf2r* promoter, thereby inhibiting RNA polymerase II recruitment [21]. Alternatively, transcriptional interference can occur over enhancer domains when so-called Enhancer Occlusion Transcripts (EOTrs) cover these regions [22]. In the current study, we identified a previously unannotated antisense transcript (unannotated AS). The maternally biased expression of this unannotated AS might occur through transcriptional interference of its intronic enhancer domains, which are occupied by *ZNF791-like* transcripts acting as EOTrs, or through the latter mechanism of RNAi, mediated by the paternally expressed *ZNF791* transcripts. These processes occur without overlapping the promoter region of unannotated AS, indicating potential transcriptional interference at an enhancer or RNAi-mediated silencing of the unannotated AS. Of note, this novel unannotated AS transcript was not observed in any other analyzed domesticated species (Figs 5B, S13A, S13B and S19) except in pigs, specifically in pig embryos, oocytes, and liver tissues (Figs 2, 5A, S13C and S13E). Whether this unannotated transcript arose during the speciation of *Sus scrofa* remains an open question. Further in-depth transcript profiling across various tissues from diverse domesticated animals is needed to determine its evolutionary origin and status of imprinting.

During oogenesis, RNA transcription, initiating from an upstream transcription start site and extending across intragenic CpG island promoters, is associated with the establishment of DNA methylation imprints at these CGI promoters, which are situated within actively transcribed regions [18,23–25] (S18 Fig). Accordingly, in our analyses on pigs and cattle, we discovered that the transcription of long *ZNF791* mRNA in oocytes overlaps with intragenic CGI-spanning gDMRs, indicating acquisition of the upstream transcript over the existing CGI that resulted in establishing maternal imprints. Additionally, we identified upstream alternative promoters that are active in porcine and bovine oocytes, supporting the transcription of the long overlapping *ZNF791* transcript. These upstream promoters overlapped with transposable elements (TEs) as observed in the mouse *Impact* locus [18]. It has been suggested that the mechanisms underlying genomic imprinting in therian genomes may have evolved from an epigenetic defense system against integrated TEs [26,27]. Furthermore, transcripts initiated from species-specific LTRs have been implicated in the establishment of DNA methylation in oocytes [18,24]. The LTR52, identified in the *ZNF791* locus, may have undergone mutations

during evolution but might act as artiodactyl-specific alternative promoters that induce transcription and CGI methylation. All *ZNF791-like* and *ZNF791* genes in analyzed livestock are transcribed in the reverse direction relative to their respective genomes. A Dfam database search showed that, except in goats, LTRs are located on the + strand, suggesting that they may act as antisense promoters (S16 Fig). This is similar to the human *SCIN* locus, where the gene is transcribed in the forward direction and an upstream LTR is located on the–strand [18], as well as other loci with opposite transcriptional orientations, as listed in S1 Table of reference [18]. Moreover, in mouse oocytes, transcribed regions are enriched with H3K36me3, which mediates *de novo* DNA methylation, thereby leading to the establishment of maternal imprints [16]. Conversely, H3K36me2 is broadly deposited in intergenic regions, mediating moderate levels of DNA methylation [17,28]. However, in porcine and bovine oocytes, both H3K36me2 and H3K36me3 modifications have been observed to be enriched within transcribed regions and showed a high correlation with DNA methylation [29]. Although these modifications were somewhat more widespread in porcine oocytes than in bovine, their distribution appeared to overlap supporting their conserved role in methylation processes. During post-fertilization, the *ZNF791* DMRs/ICRs might be selectively preserved by being protected against the genome-wide demethylation occurring before embryo implantation, potentially through the involvement of ZFP57 and/or ZNF445 as key factors [14,30,31]. Among these, ZNF445 is highly conserved across domesticated animals (S23A and S23B Fig) and is expressed starting from the stage of porcine and bovine oocytes [14], whereas ZFP57 is less conserved (S23C and S23D Fig), absent in porcine and bovine oocytes, and gradually increases in subsequent embryonic stages [14]. Notably, the imprinted *ZNF791* DMRs of pigs and cattle, particularly in CGIs, contain the TGCCGC motif, which is recognized as the consensus binding motif for ZFP57 [30] and potentially for ZNF445 [31] (S23E and S23F Fig; S6 Table). This motif is also present in CGIs within the imprinted *ZNF791* DMRs of other domestic species (S23E and S23F Fig; S6 Table). On the other hand, these *ZNF791* CGIs do not contain the two previously predicted ZNF445 motifs (ZNF445 TO and ZNF445 TR) [32] (S23E Fig and S6 Table).

Over the past three decades, more than 200 imprinted genes have been identified in both mice and humans, with 63 of these genes being common to both species and the majority being species-specific, underscoring both evolutionary conservation and divergence [11]. In domestic pigs, approximately 40 genes have been identified as imprinted, with about 30 of these being common when compared to mice and humans and about a dozen being unique to pigs, according to a previous publication [7] and geneimprint.com (accessed Feb 2024). This indicates that the current catalog of porcine imprinted genes may be insufficient to fully understand the complexity and variability of the pig epigenome. Recently, the number of imprinted genes in marsupial mammals was estimated to be around 60 [33] suggesting divergence of the imprinting processes across Theria. This divergence can be attributed to molecular factors, as the maternal gDMRs for *ZNF791* and *Impact* appear to have been established at irregular time points during mammalian evolution. Moreover, *de novo* DNA methylation-independent non-canonical imprinting discovered in rodents, particularly in the placenta, provides further targets for investigating the conservation and divergence of extraembryonic imprinting [34–38]. In domestic animals, parent-of-origin effects on body size, body weight, and feed efficiency, which are transmitted either paternally or maternally, have been reported in cross-bred species [7]. The potential impact of genomic imprinting on these agronomically important traits can also be evaluated on a large scale using experimental livestock populations [39]. This approach is grounded in the parental conflict theory, which suggests that paternally expressed genes (PEGs) tend to promote growth, while maternally expressed genes (MEGs) favor resource preservation and reduced growth [40]. As mentioned, a SNP is located at a putative promoter of the unannotated AS, which may regulate expression of unannotated AS and consequently affect the overlapping

*ZNF791-like* expression through transcriptional interference or RNAi-mediated silencing mechanisms. Because this SNP was associated with RFI, according to the pig QTLdb based on a recent GWAS [41], and imprinted genes are dosage-sensitive [42], further research on RFI and genomic imprinting in this locus is needed to investigate the functional relevance of the unannotated AS gene and the role of this imprinted gene cluster. Overall, our findings convey molecular and evolutionary insights into species-specific genomic imprinting in the *ZFN791* locus and may benefit future investigations on genetic selection of livestock.

## Methods

### Ethics statement

All animal procedures were conducted All animal procedures were conducted in accordance with guidelines approved by the Institutional Animal Care and Use Committee (IACUC) of the National Institute of Animal Science, Rural Development Administration (RDA) of Korea (NIAS2015-670).

### Sample preparation

The method of parthenogenesis has been described in our previous report [43]. Briefly, ovaries of Landrace × Yorkshire × Duroc (LYD) pigs were obtained from a slaughterhouse. Cumulus-oocyte complexes (COCs) were collected and washed in Tyrode's lactate-HEPES-polyvinyl alcohol. Oocytes with several layers of cumulus cells were selected and washed three times in TCM-199 based medium (GIBCO, Grand Island, NY, USA). For *in vitro* maturation (IVM), 50 COCs were transferred into 500 μL of maturation medium in four-well dishes and matured for 40 h at 38.5°C in an incubator containing 5% $CO_2$.

The cumulus cells were removed, and oocytes having the first polar body were selected and placed in a fusion chamber with 250 μm diameter wire electrodes (BLS, Budapest, Hungary) covered with 0.3 mol/L mannitol solution containing 0.1 mM $MgSO_4$, 1.0 mM $CaCl_2$, and 0.5 mM HEPES. To achieve fusion, two DC pulses (1 sec interval) of 1.2 kV/cm were applied for 30 μs using an LF101 Electro Cell Fusion Generator (Nepa Gene Co., Ltd., Chiba, Japan). After a 2-h stabilization period, the parthenogenetically activated (PA) embryos were transferred into the oviducts of two LYD surrogate gilts, each aged 12 months on the estrus phase, for further development.

To generate fertilized control (CN) embryos, two LYD gilts were naturally mated with boars twice with a 6-h interval during their natural heat period at the onset of estrus. Ten embryos from each of the parthenogenetically activated and control groups were recovered from two euthanized gilts 21 days after the onset of estrus, with half collected from each dam. This timing prevented the abnormal morphological changes in parthenogenetic embryos that typically appear around day 30, while the sizes of parthenogenetic embryos were smaller than that of controls prior to day 30 [44] (S1 Fig). Additionally, our previous histological assays showed that PA embryos developed with normal morphologies by day 26, forming major organs and tissues comparable to control embryos, although PA embryos were smaller than the control [45]. Embryos with intact morphology from both the PA and CN groups were used for sequencing. The recovery process involved sectioning the reproductive tracts, isolating the placenta from the uterus, and separating embryos from the surrounding placenta, and subsequently the samples were stored in liquid nitrogen until further use.

### Whole-genome bisulfite sequencing and downstream analyses

Genomic DNA was isolated from the whole embryos (two biological replicates for each of the PA and CN groups) and fragmented. The fragmented DNA (200 ng) underwent bisulfite

conversion using the EZ DNA Methylation-Gold Kit (Zymo Research, Irvine, CA, USA). To construct the DNA library, 1 ng of DNA was processed with the Accel-NGS Methyl-Seq DNA Library Kit (Swift Biosciences, Inc. Ann Arbor, MI, USA). PCR was conducted with adapter primers and Diastar EF-Taq DNA polymerase (Solgent, Daejeon, Korea) with thermal cycling conditions of a 3 m denaturation at 95˚C, followed by 10 cycles of 30 s at 95˚C, 30 s at 60˚C, and 30 s at 72˚C, and a final 5 m extension at 72˚C. Following bead-based purification, the DNA library was sequenced to produce paired-end reads of 151 nucleotides using the HiSeqX system operated by Macrogen Inc. (Seoul, Korea).

Read quality was evaluated using FastQC (v0.12.1). The raw WGBS reads were trimmed and filtered, using the default parameters of Trim Galore (v0.12.1) to remove adapters and short reads less than 20bp, except for additional trimming to eliminate up to 18 bp of low complexity sequence tags, introduced during the library preparation, from the 3′ end of R1 (–three_prime_clip_R1 18) and from the 5′ end of R2 (–clip_R2 18). The resulting cleaned reads (approximately 420 million pairs for PA1, 425 million pairs for PA2, 431 million pairs for CN1, and 433 million pairs for CN2), were mapped to the pig reference genome (Sscrofa11.1/susScr11) using the Bismark aligner (v0.22.3) with default parameters [46]. Mapping efficiency ranged from 79.6% to 81.3% (S1 Table). Deduplication was performed with deduplicate_bismark, removing PCR-based duplications that comprised between 13.00% and 15.21% of the mapped pairs. Methylation percentages for cytosines in the CpG context were determined using bismark_methylation_extractor. CpG sites covered by at least three reads and with a median of mean CpG coverages greater than 10 for all analyzed transcripts in each sample (see S2 Table, footnote) were further analyzed.

Genomic coordinates of all 2,900 porcine zinc finger transcripts from autosomes (chr1 ~ chr18) were extracted from the GFF file for the pig reference genome, excluding unplaced scaffolds (chrUn), sex chromosomes (chrX and chrY), and the mitochondrial chromosome. Profile plots and heatmaps for the transcript bodies (TSS ~ TES) and ± 10 kb regions (–10 kb from TSS and +10 kb from TES) were generated using computeMatrix, plotProfile, and plotHeatmap from deepTools (v3.5.5) [47]. A heatmap for mean DNA methylation levels of all CpG sites within the putative promoter regions (±2 kb from TSS) of selected transcripts was generated using heatmap.2 of the gplots package (v3.1.3.1). For this transcript selection, all 2,900 zinc finger transcripts were initially sorted based on their methylation ratios, defined as the mean methylation level in PA embryos divided by the mean methylation level in CN embryos. From this sorted list, the top 88 transcripts having orthologous loci in humans were identified and re-sorted based on their mean methylation levels in the first PA embryo.

Using the BSmooth method from bsseq (v1.38.0) with default parameters [48], we initially obtained methylation profile curves and candidate DMRs within the putative promoter regions. To avoid possible false positives, we further applied additional procedures using the metilene software (v0.2–8) [49]. For the metilene analysis, we employed default parameters, which include a maximum distance between CpG dinucleotides (-M) of 300, a minimum number of CpGs (-m) of 10, and a mode of operation (-f) of *de novo*. Moreover, we specified a more stringent minimum mean methylation difference (-d) of 0.2 and applied multiple testing correction (-c) with a false discovery rate (FDR) adjustment. We designated regions with an FDR < 0.05 as DMRs. DNA methylation ratios were plotted as histograms using the R package Gviz (v1.44.0) [50].

## Analyses of hemi-methylation

For the analysis of WGBS reads overlapping with partially methylated regions, we utilized MethylDackel v0.6.1 [51] to extract the names and locations of individual reads, count the

CpG sites on each read, and calculate the methylation percentage (S4 Table). We retained only those WGBS reads containing at least three CpGs sites, referred to as 'qualified reads', and partially methylated regions supported by more than 30 such qualified reads for further analyses, based on a previously described SNP-free method for outbred animals [15]. From the percentage of methylated CpGs (mCpGs) for each qualified read, hypomethylated reads (0–20%) and hypermethylated reads (80–100%) were determined. After calculating the percentages of these hypo- and hypermethylated reads among all reads in the partially methylated regions, we plotted the hemi-methylation tendency of these regions, defined as regions with more than or equal to 30% of hypomethylated reads and more than or equal to 30% of hypermethylated reads (S4 Table). For consecutive CpG sites within the partially methylated regions, CpG methylation pattern plots, referenced from Figure S2A in Xie et al. (2012) [52], were drawn using a web-based tool, QUMA (quantification tool for methylation analysis) (S4C–S4E Fig) [53].

## RNA sequencing and data processing

Total RNA was isolated from the whole embryos (three biological replicates for the PA group and six biological replicates for the CN group) using TRIzol reagent (Sigma-Aldrich, USA). The RNA samples were treated with DNase I to avoid genomic DNA contamination and electrophoresed in 1.2% agarose gels to evaluate RNA integrity, which was confirmed by a 28S/18S rRNA ratio greater than 2 and an RNA integrity number (RIN) greater than 7, using an Agilent 2100 BioAnalyzer. The concentration of RNA was assessed using the ratios of A260/A280 and A260/A230 (1.8–2.0). To select poly-A tails and construct cDNA libraries, 1 µg of total RNA was used with the TruSeq RNA Sample Prep Kit v.2 (Illumina, San Diego, CA, USA). The cDNA libraries were quantified by quantitative Real-Time PCR (qPCR). The Illumina HiSeq2500 RNA-seq platform was used to sequence the library products to produce 101 nucleotide paired-end reads.

Read quality was assessed using FastQC (v0.12.1). To remove adapters and low-quality reads from the raw RNA-seq data, we utilized Trimmomatic (v0.39), applying parameters LEADING:3 TRAILING:3 SLIDINGWINDOW:4:15 MINLEN:36 [54]. This process yielded approximately 27 to 43 million read pairs for the PA and CN embryo samples. Using the STAR aligner (v2.7.9a), the cleaned reads were aligned to the reference genome (Sscrofa11.1/susScr11) with default parameters [55], resulting in 96.0% to 96.5% of the reads being uniquely mapped (S1 Table). These reads were retrieved with SAMtools [56] and 20.0% to 25.0% of them were subsequently deduplicated using Picard MarkDuplicates for further analyses. BAM files were normalized to generate BigWig files using bamCoverage from deepTools (v3.5.5) [47], with options—binSize 10 and—smoothLength 15. These files were then visualized using Gviz (v1.44.0) [50]. Transcript quantification was performed using Salmon (v1.3.0) in mapping-based mode [57], calculating transcript per million (TPM) values.

## Data mining and processing

Our generated data, along with publicly available data downloaded from the NCBI Gene Expression Omnibus (GEO) and other resources, are listed in S3 Table. WGBS and RNA-seq data were processed using the procedures described earlier in this report, while ChIP-seq, whole genome sequencing (WGS), and whole exome sequencing (WES) data were processed as previously outlined in our report [58]. Reference genomes used in this study include GRCh38.p14/hg38 (human), macFas5 (crab-eating macaque), panTro6 (chimpanzee), rheMac8 (rhesus monkey), Asia_NLE_v1 (gibbon), GRCm39/mm39 (mouse), rat (rn7), equCab3 (horse), canFam3 (dog), susScr11 (pig), bosTau9 (cow), oriAri4 (sheep) and CHIR_1.0 (goat).

The VCF files used for SNP analyses are listed in S3 Table. For VCF files based on a previous genome, the Picard LiftoverVcf (v2.23.8) was used to convert them to match the reference genome mentioned above (S3 Table).

### LTR search

When there are multiple *ZNF791* orthologs, we focused on the *ZNF791* gene immediately downstream (3′) of the *MAN2B1* gene across species. We extracted sequences of the promoter regions of the *ZNF791* gene and conducted searches for transposable elements (TEs) in the Dfam database [59] (release 3.8; accessed on December 28, 2023), which utilizes profile hidden Markov models (HMMs), as detailed in S16 Fig. For each species analyzed, the search was parameterized with the—species option, while employing the default gathering (GA) threshold via—cut_ga, the Dfam curated threshold, to ensure specificity to repetitive element families. For dogs, however, which did not yield significant matches under the GA threshold, we adapted our approach by utilizing an E-value threshold (-E 0.1) to identify potential LTRs.

### Phylogenetic analyses

For phylogenetic analyses, LTR-derived sequences from domesticated animal species and the consensus LTR52 sequence (DF000000543.4) from the Dfam database [59] (accessed December 28, 2023) were aligned using MUSCLE for multiple sequence alignment, as implemented in the MEGA11 software [60]. Using MEGA11, the genetic distances were calculated by the Kimura 2-parameter substitution model with uniform rates among sites [61] and maximum-likelihood phylogenetic trees (500 bootstrap replications) were constructed. For another construction of phylogenetic trees with divergence times estimated by TimeTree 5, resulting Newick files were downloaded from http://www.timetree.org (accessed Feb 5, 2024) [62]. The phylogenetic trees were edited using FigTree (v.1.4.4) [63].

### Motif analyses

*De novo* motif discovery was conducted using the MEME tool from the MEME Suite [64], and the discovered motifs were compared with the JASPAR2022 CORE vertebrates non-redundant database using the TOMTOM tool. Also, motif scanning was performed using the FIMO tool with transcription factor binding sites (TFBSs) data downloaded from the AnimalTFDB v4.0 database [64]. Motif logos were drawn using the R package universalmotif (v1.20.0).

### Statistical analyses

For DEG analysis, the output files generated by Salmon were imported into R and analyzed using the DESeq2 package (v.1.28.1) [65]. DEGs were identified based on two criteria: an absolute log2-fold change greater than 1 and an FDR below 0.05. To compare mean expression levels at the transcript level between two unrelated groups, specifically PA and CN embryos, the Welch two sample t-test, implemented in R, was used.

## Supporting information

**S1 Fig. Measurement of porcine embryos.**
(PDF)

**S2 Fig. CpG coverages and a positive control for the pig imprinted gene.**
(PDF)

**S3 Fig. Identification of expressed transcripts from RNA-seq data of porcine PA and CN embryos.**
(PDF)

**S4 Fig. Partial DNA methylation at the *ZNF791* locus in pigs downstream of the *MAN2B1* gene.**
(PDF)

**S5 Fig. Partial DNA methylation at the *ZNF791* locus in cattle downstream of the *MAN2B1* gene.**
(PDF)

**S6 Fig. Partial DNA methylation at the *ZNF791* locus in sheep downstream of the *MAN2B1* gene.**
(PDF)

**S7 Fig. Partial DNA methylation at the *ZNF791* locus in horses downstream of the *MAN2B1* gene.**
(PDF)

**S8 Fig. Partial DNA methylation at the *ZNF791* locus in goat downstream of the *MAN2B1* gene.**
(PDF)

**S9 Fig. Partial DNA methylation at the *ZNF791* locus in dogs downstream of the *MAN2B1* gene.**
(PDF)

**S10 Fig. DNA methylation at the *ZNF791* locus in pig embryos downstream of the *MAN2B1* gene.**
(PDF)

**S11 Fig. Human lung exome from WES and RNA-seq.**
(PDF)

**S12 Fig. Expressed *ZNF791* transcripts in humans, primates, and mice.**
(PDF)

**S13 Fig. Expressed *ZNF791* transcripts in dogs, cattle, and pigs.**
(PDF)

**S14 Fig. Expressed transcripts at the *ZNF791* locus in pig oocytes.**
(PDF)

**S15 Fig. An expressed unannotated transcript at the *ZNF791* locus in pig oocytes.**
(PDF)

**S16 Fig. Dfam database search process.**
(PDF)

**S17 Fig. Expressed transcripts at the *ZNF791* locus in cow oocytes.**
(PDF)

**S18 Fig. LTR-initiated transcription and establishment of methylation imprint in mouse oocytes.**
(PDF)

**S19 Fig. Expressed *ZNF791* transcripts in sheep, horses, and goats.**
(PDF)

**S20 Fig. Multiple sequence alignment of LTRs.**
(PDF)

**S21 Fig. Motif discovery and comparison.**
(PDF)

**S22 Fig. QTL analysis for the upstream of the unannotated antisense transcript.**
(PDF)

**S23 Fig. Sequence and binding motif analysis.**
(PDF)

**S1 Table. Summary of data metrics.**
(XLSX)

**S2 Table. Identification of candidate DMR-associated transcripts.**
(XLSX)

**S3 Table. Deposited data.**
(XLSX)

**S4 Table. Analyses of partially methylated regions.**
(XLSX)

**S5 Table. Individual-matched analyses of genomic DNA and RNA-seq read counts on informative SNPs.**
(XLSX)

**S6 Table. FIMO analyses of binding motifs.**
(XLSX)

## Acknowledgments

We are thankful to Ms. Michelle Milligan for proofreading this manuscript.

## Author Contributions

**Conceptualization:** Jinsoo Ahn, Seongsoo Hwang, Kichoon Lee.

**Data curation:** Jinsoo Ahn.

**Formal analysis:** Jinsoo Ahn, In-Sul Hwang, Mi-Ryung Park, Milca Rosa-Velazquez.

**Funding acquisition:** Kichoon Lee.

**Investigation:** Jinsoo Ahn, In-Sul Hwang, Mi-Ryung Park, Milca Rosa-Velazquez, Alejandro E. Relling.

**Methodology:** Jinsoo Ahn, In-Sul Hwang, Mi-Ryung Park, In-Cheol Cho.

**Resources:** In-Sul Hwang, In-Cheol Cho, Alejandro E. Relling.

**Software:** Jinsoo Ahn.

**Supervision:** Seongsoo Hwang, Kichoon Lee.

**Validation:** Jinsoo Ahn, In-Sul Hwang, Mi-Ryung Park, Seongsoo Hwang, Kichoon Lee.

**Visualization:** Jinsoo Ahn.

**Writing – original draft:** Jinsoo Ahn, Kichoon Lee.

**Writing – review & editing:** Jinsoo Ahn, Kichoon Lee.

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
