## [Decision Letter · Decision Letter 0]

16 Sep 2024

Dear Dr Kichoon Lee,

Thank you very much for submitting your Research Article entitled 'Lineage-specific genomic imprinting in the ZNF791 locus' to PLOS Genetics.

The manuscript was fully evaluated at the editorial level and by independent peer reviewers. All three reviewers appreciated the interest and quality of your study, and commented on the originality of the findings relative to the domesticated species studied.  However, they also raised several minor and major concerns about the presentation and description of the experiments, including about how many and which precise embryo and RNA samples were studied and how these are correlated.  Besides other minor points, questions were raised also about the identification of the strand-specific transcripts, and about the depth of the DNA methylation data.  The rationale of the study was mentioned as well and would require further explanation.

We invite you to prepare a revision and therefore ask you to modify the manuscript according to the review recommendations. Your revisions should address the specific points made by each reviewer.

To resubmit, log into your Editorial Manager account and select the option 'Revise Submission' in the 'Submissions Needing Revision' folder.

Yours sincerely,

Robert Feil, PhD

Guest Editor

PLOS Genetics

John Greally

Section Editor

PLOS Genetics

Reviewer's Responses to Questions

**Comments to the Authors:**

Reviewer #1: Synopsis

In this study, the authors compare parthenogenetic and normal pig embryos to identify

new imprinted genes. By limiting their search for Zn finger protein genes with human

orthologs, they identify maternal gDMRS at the PLAGL1 and ZNF791-like genes. They

show expression data suggesting that the ZNF791-like gene is exclusively expressed

from the paternal allele. They show that the differential methylation at the ZNF791 gene

is conserved in domesticated animals (pig, cow, goat, sheep, horse and dog) and

absent in primates and rodents. Using expressed SNPs, they confirm monoallelic

expression in species with the gametic DMR. Analysis of published data from pig and

cow suggest the implication of an LTR element, active as a promoter in oocyte, in

transcription across the DMR and de novo DNA methylation. This was not observed in

human and mouse oocytes. Evidence for a similar LTR-driven mechanism is presented

for other imprinted species. Overall, the study present good evidence that imprinting at

ZNF791 in domesticated species correlates with integration of an LTR element at the

locus.

The paper is generally well written and the interpretation and conclusions are generally

well supported by the new data presented as well as the mining of published data and

their analyses by the authors. However, the manuscript could be improved by

addressing the following comments.

Comments

1. The authors limited their gDMR search to 642 zinc finger genes but do no provide

any justification for this choice. Why only those genes were queried?

2. Similarly, from the top differentially methylated ZNF genes only those with human

orthologs were considered. Why? The authors need to justify the basis for this

selection.

3. For the profiles presented in Fig.1C, the authors should annotate the X axes and

show the position of the TSS and exons. The bottom tick marks presumably show

the position of CpGs; this should be clarified on the figure.

4. Line 139-140 - “and the corresponding paternal expression pattern was exclusively

observed in the control embryos.” The authors refer to this expression data but have

not yet mentioned that they have conducted RNA-seq analysis in the paper. This

should be presented first.

5. Although the author suggest the formation of an antisense transcript, no strand-

specific data is presented. Is the RNA-seq data stranded? If not, the proper

annotation of this transcript would require usage of a strand-specific RT-PCR

experiment.

6. Lines 183-186 - Although consistent with imprinting, differential methylation between

oocyte and sperm is not “indicating that maternal silencing and paternal monoallelic

transcriptional initiation of ZNF791”. This would need to be confirmed by analysis of

expression. Nevertheless, the DNA methylation data in somatic tissues do show that

the gametic DMR is maintained and lend further support for imprinting of the locus.

This needs to be clarified.

7. Line 239-241 - The statement about the roles of H3K39 methylation in de novo DNA

methylation in oocytes needs to refer to these two publications:

Xu, Q. et al. SETD2 regulates the maternal epigenome, genomic imprinting and

embryonic development. Nature Genetics 51, 844–856 (2019).

Yano, S. et al. Histone H3K36me2 and H3K36me3 form a chromatin platform

essential for DNMT3A-dependent DNA methylation in mouse oocytes. Nat Commun

13, 4440 (2022).

8. In the section starting on line 259, the authors make a parallel between what they

observed at ZNF791-like genes and at the mouse Impact locus. In fact, two different

mouse genes have been confirmed to acquired DNA methylation in oocytes as a

consequence of transcription from an upstream LTR promoter, Impact and Slc38a4.

This should be clarified, with reference to this publication, which is only mentioned in

sFig.18 but not cited in the paper nor included in the bibliography:

Bogutz, A. B. et al. Evolution of imprinting via lineage-specific insertion of retroviral

promoters. Nat Commun 10, 5674 (2019).

This also applies to the sentences starting on line 319 and 372, where the same

reference should be included.

9. Lines 314-318 - It is not clear to this reviewer why the analysis presented is “contrary

to” to emergence of imprinting after the split from monotremes. In fact, it is

consistent with this model, since imprinting of ZNF791 is more recent. This needs to

be fixed in the text.

10. Line 363 - Although an RNAi silencing may occur here, the downregulation of the

paternal allele of the antisense transcript could also be mediated via a transcriptional

interference mechanism implicating enhancer occlusion by ZNF971, as proposed at

several loci in this publication:

Pande, A., Makalowski, W., Brosius, J. & Raabe, C. A. Enhancer occlusion

transcripts regulate the activity of human enhancer domains via transcriptional

interference: a computational perspective. Nucleic Acids Res. 48, 3435–3454

(2020).

11. Line 373 - The original suggestion of a link between genomic imprinting and a

defense mechanism comes from this publication by Denise Barlow, which predates

reference #16 given by the authors:

Barlow, D. P. Methylation and Imprinting: from Host Defense to Gene Regulation?

Science 260, 309–310 (1993).

12.Line 375 - In addition to reference #17, the authors also need to refer here to the

Bogutz reference given above in point 8.

13.Line 378 - A reference to the Xu paper mention above (point 7) needs to be added

here.

14. Line 385 - Protection from demethylation may involve ZFP57/445 or similar DNA

methylation-dependent binding of a Zn finger protein. This should be mentioned

here. The authors should mention whether these proteins are conserved in the

domesticated species studied here and whether the imprinted ZNF791 DMRs

contain binding motifs for these proteins.

15.Line 395 - Here again, the statement about imprinting evolution is confusing. The

model is not that ALL imprinting occurred ~160 MYA ago, but rather that the first

examples of imprinting are observed at this time. The fact that lineage-specific

insertion of LTRs may have contributed to the emergence of species-specific

imprinting at different time point since then is to be expected. In fact, examples of

such events unique to the primate lineages have also been presented at 17 loci by

Bogutz et al.

16.I do not recall this being mentioned in the paper, but the authors should clarify

whether the documented LTR promoters are indeed transcriptionally oriented

towards ZNF791.

Minor comments

1. The developmental stage of the embryos collected (21 days after the onset of

estrus) should be mentioned in sFig. 1.

2. Line 148, change “was explored to reveal the imprinting status” to “was explored to

study its imprinting status”, to reflect that the imprinted expression was not yet

shown in the paper.

3. sFig.2 - “Non-expressed PLAGL1 transcripts are covered with white shading for

visual distinction.” It seems that the author actually mean transcripts not expressed

in their embryonic samples. This should be clarified.

4. In sFig.3, the position of the CpG islands should be presented.

5. Line 202 - This is a reference to Fig. 3G, not 3F. Otherwise, there is no reference to

3F in the text.

Reviewer #2: In this manuscript, the authors generate transcriptome and methylome datasets in parthenogenetic porcine embryos to identify potentially novel imprinted genes. They use this data and previously published datasets from other species to reveal for the first time the species-specific imprinting of ZNF791 in domesticated animals but not human and mouse. The authors also convincingly show that the imprinting of ZNF791 correlates with the insertion of an LTR element that drives upstream transcription in the oocyte.

The manuscript is clear and convincing. To my opinion, it represents a nice and important contribution to the field of genomic imprinting that is of general interest for the scientific community.

I have only minor comments that need to be addressed for publication, as detailed below.

Comments:

-What is the rationale for initially focusing the analysis on zinc finger protein genes? Are these prone to imprinting in other species? This could be better explained in the introduction.

-The WGBS datasets seem to be of good quality with a high number of reads. Can the authors provide some information on the sequencing depth and genome coverage?

-Figure 1B “Based on the methylation ratios between PA and CN embryos, the top 88 porcine zinc finger transcripts with orthologous loci in humans were identified”.

Please be more specific. How where the top 88 candidate transcripts identified? Are these the 88 transcripts with the most methylation difference or only the ones with orthologous loci in humans? What cut off was used ? this is not clear.

-Figure 1C: In addition to the smoothed methylation profiles, can the authors also show the real methylation at CpG resolution for this region (a zoom in view of the DMR from data in Figure 2C)?

-Lines 159-160 “In contrast, the CGI regulates transcription of the ENSSSCG00000033388 gene was unmethylated in both PA and CN embryos (Fig. 2C)”. Please correct, there seem to be a grammatical mistake.

-Lines 180-186: in the text, the authors directly correlate hypomethylation with transcription initiation of the gene in gametes. I would tone this down because there is no expression data. The authors should only mention lack or conservation of imprinted gametic methylation.

-The analysis of individual reads to confirm the co-occurrence of fully methylated and hypomethylated sequences in DMRs (Figure 3E-F) is appreciated. However, I don’t understand how these tendency figures were generated from the real WGBS datasets. More details need to be provided to make sure that these profiles rigorously reflect the reality of all the WGBS reads.

-Have the authors tried to use SNPs in the DMR region to demonstrate parental-specific methylation in the existing WGBS datasets of somatic tissues from domestic animals?

-Lines 200-204: Figures 3F and 3G are not correctly cited in the text.

-Line 516: I think “Supplementary Table 2” should be replaced by “Supplementary Table 3”.

Reviewer #3: The review is uploaded as an attachment

**Have all data underlying the figures and results presented in the manuscript been provided?**

Reviewer #1: Yes

Reviewer #2: Yes

Reviewer #3: Yes

PLOS authors have the option to publish the peer review history of their article (what does this mean?). If published, this will include your full peer review and any attached files.

Reviewer #1: **Yes: **Louis Lefebvre

Reviewer #2: No

Reviewer #3: No

---

## [Decision Letter · Decision Letter 1]

11 Nov 2024

PGENETICS-D-24-00906R1Lineage-specific genomic imprinting at the ZNF791 locusPLOS Genetics Dear Dr. Lee, Thank you for submitting your revised manuscript to PLOS Genetics. As you will see, the three reviewers are satisfied with the extensive revision and are in principle in favour of publication.  However, reviewer 1 still has minor suggestions for changes in the text, which I am confident you will be able to make.    From my side, I would like to suggest a small change in the Title to better reflect what this study is about: 'Evolutionary lineage-specific genomic imprinting at the ZNF791 gene'.   In fact, many studies have considered developmenal lineage-specific imprinting, and to not confuse the reader and highlight the specificity of your study, it would be good to include the word 'evolutionary' into the title as suggested. Please submit your revised manuscript at an early convenience and no later than in 30 days.  Please include the following items when submitting your revised manuscript:*
A rebuttal letter that responds to each point raised by the editor and reviewer(s). You should upload this letter as a separate file labeled 'Response to Reviewers'. This file does not need to include responses to formatting updates and technical items listed in the 'Journal Requirements' section below.*
A marked-up copy of your manuscript that highlights changes made to the original version. You should upload this as a separate file labeled 'Revised Manuscript with Track Changes'.*
An unmarked version of your revised paper without tracked changes. You should upload this as a separate file labeled 'Manuscript'. If you would like to make changes to your financial disclosure, competing interests statement, or data availability statement, please make these updates within the submission form at the time of resubmission. Guidelines for resubmitting your figure files are available below the reviewer comments at the end of this letter. We look forward to receiving your revised manuscript shortly. Kind regards, Robert Feil, PhDGuest EditorPLOS Genetics John GreallySection EditorPLOS Genetics Aimée DudleyEditor-in-ChiefPLOS Genetics Anne GorielyEditor-in-ChiefPLOS Genetics **Journal Requirements:** **Additional Editor Comments (if provided):****Reviewers' comments:** Reviewer's Responses to Questions

**Comments to the Authors:**

Reviewer #1: I thank the authors for addressing my comments appropriately. I think this revised version is much improved.

However, two minor points still need to be addressed.

Lines 401-403 of the revised manuscript: “Alternatively, transcriptional interference can occur over enhancer domains when so-called Enhancer Occlusion Transcripts (EOTrs) occupy these regions.”

“Occupy” should be changed to “cover”

In response to my comment #14, the authors added lines 442-445 to discuss the implication of ZFP57/ZNF445 in DNA methylation maintenance at the ZNF791 DMR during preimplantation stages. However, they still need to specifically mention whether the imprinted ZNF791 DMR

contains binding motifs for these proteins.

Reviewer #2: The authors have addressed all my comments in the revised manuscript.

The work is a nice and important contribution to the field of genomic imprinting

Reviewer #3: I'd like to thank the authors and congratulate them for taking all the comments into account and for their answers. The manuscript was already good, it's even better now and the results are very interesting for the scientific community!

**Have all data underlying the figures and results presented in the manuscript been provided?**

Reviewer #1: Yes

Reviewer #2: Yes

Reviewer #3: Yes

PLOS authors have the option to publish the peer review history of their article (what does this mean?). If published, this will include your full peer review and any attached files.

Reviewer #1: No

Reviewer #2: No

Reviewer #3: **Yes: **Julie Demars

 **Figure resubmission:** While revising your submission, please upload your figure files to the Preflight Analysis and Conversion Engine (PACE) digital diagnostic tool, https://pacev2.apexcovantage.com/. PACE helps ensure that figures meet PLOS requirements. To use PACE, you must first register as a user. Registration is free. Then, login and navigate to the UPLOAD tab, where you will find detailed instructions on how to use the tool. If you encounter any issues or have any questions when using PACE, please email PLOS at figures@plos.org. Please note that Supporting Information files do not need this step. If there are other versions of figure files still present in your submission file inventory at resubmission, please replace them with the PACE-processed versions. **Reproducibility:** To enhance the reproducibility of your results, we recommend that authors deposit laboratory protocols in protocols.io, where a protocol can be assigned its own identifier (DOI) such that it can be cited independently in the future. Additionally, PLOS ONE offers an option to publish peer-reviewed clinical study protocols. Read more information on sharing protocols at https://plos.org/protocols?utm_medium=editorial-email&utm_source=authorletters&utm_campaign=protocols

---

## [Editor Report · Decision Letter 2]

9 Dec 2024

Dear Dr Lee,

We are pleased to inform you that your manuscript entitled "Evolutionary lineage-specific genomic imprinting at the ZNF791 locus" has been editorially accepted for publication in PLOS Genetics. Congratulations!

Yours sincerely,

Robert

Robert Feil, PhD

Guest Editor

PLOS Genetics

John Greally

Section Editor

PLOS Genetics

Aimée Dudley

Editor-in-Chief

PLOS Genetics

Anne Goriely

Editor-in-Chief

PLOS Genetics

Comments from the reviewers (if applicable):

**Data Deposition**

http://datadryad.org/submit?journalID=pgenetics&manu=PGENETICS-D-24-00906R2

**Press Queries**

---

## [Editor Report · Acceptance letter]

27 Dec 2024

PGENETICS-D-24-00906R2 

Evolutionary lineage-specific genomic imprinting at the ZNF791 locus 

Dear Dr Lee, 

We are pleased to inform you that your manuscript entitled "Evolutionary lineage-specific genomic imprinting at the ZNF791 locus" has been formally accepted for publication in PLOS Genetics! Your manuscript is now with our production department and you will be notified of the publication date in due course.

With kind regards,

Zsofia Freund

PLOS Genetics

On behalf of:
